

**Development of an instrument for direct ozone production rate measurements: Measurement**
**reliability and current limitations**
Sofia Sklaveniti[1,2], Nadine Locoge[1], Philip S. Stevens[2,3], Ezra Wood[4,5], Shuvashish Kundu[4], Sebastien
Dusanter[1]
[1] IMT Lille Douai, Univ. Lille, SAGE - Département Sciences de l'Atmosphère et Génie de
l'Environnement, 59000 Lille, France
[2] School of Public and Environmental Affairs, Indiana University, Bloomington, IN 47405, USA
[3] Department of Chemistry, Indiana University, Bloomington, IN, USA
[4] Department of Chemistry, University of Massachusetts, Amherst, MA USA
[5] Department of Chemistry, Drexel University, Philadelphia, PA, USA
**Abstract**
Ground level ozone ($O_3$) is an important pollutant that affects both global climate change and regional
air quality, with the latter linked to detrimental effects on both human health and ecosystems. Ozone
is not directly emitted in the atmosphere but is formed from chemical reactions involving volatile
organic compounds (VOCs), nitrogen oxides ($NO_x = NO+NO_2$) and sunlight. The photochemical
nature of ozone makes the implementation of reduction strategies challenging and a good
understanding of its formation chemistry is fundamental in order to develop efficient strategies of
ozone reduction from mitigation measures of primary VOCs and NOx emissions.
An instrument for direct measurements of ozone production rates (OPR) was developed and deployed
in the field as part of the IRRONIC (Indiana Radical, Reactivity and OzoNe production
InterComparison) field campaign. The OPR instrument is based on the principle of the previously
published MOPS instrument (Measurement of Ozone Production Sensor) but using a different
sampling design made of quartz flow tubes and a different $O_x$ ($O_3$ and $NO_2$) conversion/detection
scheme composed of an $O_3$-to-$NO_2$ conversion unit and a Cavity Attenuated Phase Shift (CAPS) $NO_2$
monitor. Tests performed in the laboratory and in the field, together with model simulations of the



radical chemistry occurring inside the flow tubes, were used to assess (i) the reliability of the
measurement principle and (ii) potential biases associated to OPR measurements.
This publication reports the first field measurements made using this instrument to illustrate its
performance. The results showed that a photo-enhanced loss of ozone inside the sampling flow tubes
disturb the measurements. This issue needs to be solved to be able to perform accurate ambient
measurements of ozone production rates with the instrument described in this study. However, an
attempt was made to investigate the OPR sensitivity to $NO_x$ by adding NO inside the instrument. This
type of investigations does not require measuring ambient OPR but only probing the change in ozone
production when NO is added. During IRRONIC, changes in ozone production rates ranging from the
limit of detection ($3\sigma$) of 6.2 ppbv h$^{-1}$ up to 20 ppbv h$^{-1}$ were observed when 6 ppbv of NO was added
into the flow tubes.
**1    Introduction**
Ground-level ozone ($O_3$) is a primary constituent of photochemical smog that irritates the respiratory
system (WHO, 2013) and damages vegetation (Ashmore, 2005). In addition, ozone is a greenhouse
gas and an important precursor of the hydroxyl radical (OH), a key species controlling the
atmospheric oxidative capacity (Monks, 2005;Rohrer et al., 2014;Prinn, 2003). Ozone is a
photochemical pollutant formed during daytime and has an average lifetime estimated at $22\pm2$ days
(Stevenson et al., 2006), which is long enough to transport it from polluted regions to remote areas
and between continents. The local production of ozone on top of the amount advected from elsewhere
can lead to exceedances of air quality standards in urbanized areas, making ozone pollution an issue
of global concern (Akimoto, 2003).
In the troposphere, ozone can be rapidly converted to nitrogen dioxide ($NO_2$) through reaction with
nitric oxide (NO), and back to $O_3$ through $NO_2$ photolysis. This chemistry does not produce new
ozone and is known as the $O_3$-$NO_x$ PhotoStationary State (PSS), with $NO_x$ being the sum of NO and
$NO_2$. The production of new ozone is driven by the oxidation of Volatile Organic Compounds
(VOCs), which leads to the production of hydroperoxy ($HO_2$) and organic peroxy ($RO_2$) radicals. The



current understanding of tropospheric ozone chemistry indicates that new ozone is formed via
reactions of these peroxy radicals with NO, which results in the conversion of NO to $NO_2$ without
consumption of ozone (Monks, 2005;Seinfeld and Pandis, 2006).
When ozone is produced, reactions of peroxy radicals with NO also lead to the formation of OH,
which can then oxidize other molecules of VOCs to produce more peroxy radicals, and as a
consequence, more ozone. The propagation chemistry between $RO_x$ (OH, $HO_2$ and $RO_2$) radicals,
which fuels ozone production, is terminated either by $NO_x$-$RO_x$ reactions or by cross reactions of $RO_x$
radicals in $NO_x$-rich and $NO_x$-poor environments, respectively. These two types of termination
reactions lead to different regimes of ozone production referred as $NO_x$-limited or $NO_x$-saturated
when the rate of ozone production increases or decreases with $NO_x$, respectively. The turnover point
between the two regimes depends on $NO_x$ concentrations, VOC reactivity, and radical production
rates (Kleinman, 2005). Since different air quality regulations have to be implemented for the two
different regimes, i.e either $NO_x$ or VOC emission regulations, the understanding of the complex and
non-linear radical chemistry is key for the design of efficient emission control strategies.
The instantaneous ozone production rate, $p(O_3)$, can be calculated from Equation (1) as the rate of
reactions between peroxy radicals and NO. The instantaneous ozone loss rate, $l(O_3)$, can be calculated
using Equation (2), based on reaction rates for ozone photolysis, reactions of $O_3$ with $HO_x$ and
alkenes, and the reaction of OH with $NO_2$, since $NO_2$ is a reservoir molecule for $O_3$. The net ozone
production rate, $P(O_3)$, is then computed as the difference between instantaneous production and loss
rates (Eq. (3)):
$$p(O_3) = k_{HO_2+NO}[HO_2][NO] + \sum_i (k_{RO_{2,i}+NO}[RO_{2\,i}][NO]) \qquad (1)$$
$$l(O_3) =$$
$$k_{O(^1D)+H_2O}[O(^1D)][H_2O] + k_{OH+O_3}[OH][O_3] + k_{HO_2+O_3}[HO_2][O_3] +$$
$$\sum_i k_{O_3+Alkene_i}[O_3][Alkene_i] + k_{OH+NO_2}[OH][NO_2] \qquad (2)$$
$$P(O_3) = p(O_3) - l(O_3) \qquad (3)$$



Here $k_{X+Y}$ is the bimolecular reaction rate constant for the two reagents X and Y. Therefore, the
calculation of ozone production rates requires peroxy radical concentrations, either from ambient
measurements (Green et al., 2006;Liu and Zhang, 2014;Fuchs et al., 2008;Dusanter et al.,
2009a;Griffith et al., 2016) or box model outputs (Goliff et al., 2013;Stockwell et al., 2011;Saunders
et al., 2003).
In most urban and suburban environments, where concentrations of $NO_x$ are significant (10-80 ppbv),
ozone production rates can reach a few tens of ppbv $h^{-1}$ (Mao et al., 2010). In highly polluted
environments, such as Mexico City or Houston, TX, $P(O_3)$ can even exceed 100 ppbv $h^{-1}$ (Shirley et
al., 2006;Chen et al., 2010). Ozone production rates lower than 10 ppbv $h^{-1}$ have also been observed in
urban atmospheres such as Phoenix, AZ (Kleinman et al., 2002), likely due to lower initiation rates of
radicals. Ozone production is usually low in more remote areas or forested environments that are not
impacted by anthropogenic activities (less than 2-3 ppbv $h^{-1}$), due to the low $NO_x$ concentrations
(Geng et al., 2011). However, if $NO_x$ emission sources are located downwind of a forested area,
highly reactive biogenic VOCs (e.g. isoprene) can lead to an enhancement of ozone production (Geng
et al., 2011;Thornton et al., 2002).
Some studies performed in urban and suburban areas, whose objectives were to test our understanding
of the radical chemistry by contrasting measurements and model simulations of $HO_x$ concentrations,
showed that models tend to underestimate $HO_2$ for NO mixing ratios higher than a few ppbv (Ren et
al., 2013;Chen et al., 2010;Dusanter et al., 2009b;Kanaya et al., 2007;Ren et al., 2003). In contrast,
models tend to overestimate $HO_2$ in forested areas and regions characterized by large concentrations
of biogenic VOCs (Griffith et al., 2013;Mao et al., 2012;Pugh et al., 2010). Disagreements are also
present in the modeling of OH, with the models underestimating the measurements at forested
environments (Lelieveld et al., 2008;Tan et al., 2001;Whalley et al., 2011;Hofzumahaus et al.,
2009;Lu et al., 2013;Pugh et al., 2010), while the agreement may be better when colder temperatures
lead to lower concentrations of isoprene and other VOCs (Griffith et al., 2013). The discrepancies
between models and measurements question our ability to successfully measure radical species or
indicate that there are still unknowns in our understanding of the radical and ozone production



chemistry, which in turn could lead to erroneous P(O$_3$) calculations by atmospheric models. These
models are widely used for the design of air quality regulations (Rao et al., 2010; Fu et al., 2006)
based on emission control strategies. It is therefore essential to ensure that chemical mechanisms used
in atmospheric models are accurate enough to simulate the oxidative capacity of the atmosphere and
to predict both absolute rates of ozone production and the turnover point between the two ozone
production regimes.
In order to address these issues, an instrument for direct ozone production measurements (MOPS) was
developed by Cazorla and Brune (2010). The principle of MOPS is based on differential ozone
measurements between two sampling chambers made of FEP, one exposed to sunlight (referred as
sampling chamber) to get an ozone production rate inside the chamber that mimics atmospheric P(O$_3$)
and the other one covered with a UV filter (reference chamber) to suppress the radical chemistry, and
as a consequence, ozone production. The difference in ozone between the two chambers divided by
the exposure time yields the ozone production rate. However, NO$_2$ can act as a reservoir molecule for
O$_3$ due to the rapid interconversion between these two species and NO$_2$ has to be converted into O$_3$
before measuring ozone. The differential O$_x$ (O$_x$=O$_3$+NO$_2$) measurements yields P(O$_x$) values, which
represent P(O$_3$) when NO$_2$ is efficiently photolyzed during daytime.
The first version of the MOPS instrument was tested on the campus of Pennsylvania State University
in the late summer of 2008. These tests demonstrated the feasibility of the MOPS technique, as the
instrument responded to the presence of solar radiation and ozone precursors and yielded rates of
ozone production that were within a range of reasonable values (up to 10 ppbv h$^{-1}$) for this area. This
instrument was then deployed during the Study of Houston Atmospheric Radical Precursors (SHARP,
2009) (Cazorla et al., 2012). The measurements were compared to ozone production rates calculated
using measurements of HO$_2$ and NO (referred as calculated P(O$_3$)) as well as modeled radical
concentrations from a box model (referred as modeled P(O$_3$)). Measured and calculated P(O$_3$) had
similar peak values but the calculated P(O$_3$) tended to peak earlier in the morning when NO values
were higher. Measured and modeled P(O$_3$) had a similar diurnal profile, but the modeled P(O$_3$) was
only half the measured P(O$_3$). The MOPS deployment during the SHARP field campaign showed the



potential of this instrument for contributing to the understanding of the ozone-producing chemistry,
but was limited by measurement uncertainties due to potential wall effects. The heterogeneous loss of
$NO_2$ under humid conditions (RH> 50%) was reported as a main issue for this technique.
Recently, an improved version of the MOPS instrument was deployed during the NASA's
DISCOVER-AQ field campaign in 2013, in Houston, Texas (Baier et al., 2015). Wall effects were
reduced by improving the design of the sampling chambers and the airflow characteristics. The
measurements made over one month were consistent with ambient ozone observations and model-
derived $P(O_3)$ values from previous field campaigns in Houston. The authors, however, highlighted a
possible bias due to surface HONO production followed by its photolysis in the sampling chamber, as
well as unresolved ozone analyzer issues. HONO concentrations in the sampling chambers were
reported as two to five times higher than ambient values, which could cause a bias up to 5-10 ppbv h$^{-1}$
on the $P(O_3)$ measurements.
In this publication, we present the development and the characterization of an Ozone Production Rates
(OPR) instrument. The OPR instrument is based on the principle of the MOPS, using different
sampling and detection schemes. This publication describes this new instrument and its
characterization in the laboratory. An emphasis is given to the modeling of the radical chemistry
inside the sampling chambers to assess potential biases on $P(O_3)$ measurements associated to
instrumental characteristics and operating conditions. The publication also reports preliminary field
results from the Indiana Radical, Reactivity and Ozone Production Intercomparison (IRRONIC)
campaign, which highlight the current limitations of this instrument.
**2 Experimental section**
**2.1 Description of the OPR instrument**
The principle of the OPR is based on differential $O_x$ measurements between an "ambient" flow tube,
exposed to sunlight to mimic ambient photochemistry, and a "reference" flow tube, covered with an
Ultem® film (polyetherimide, 0.25 mm thick, CS Hyde Co, USA) to block wavelengths lower than
400 nm, which in turn should suppress ozone production. As mentioned above for the MOPS





instrument, the fast partitioning between $O_3$ and $NO_2$ requires measuring $O_x$ instead of $O_3$, assuming
that $P(O_3)$ is equal to $P(O_x)$ when $NO_2$ is efficiently photolyzed during daytime. $P(O_x)$ is calculated
from the difference in $O_x$ between the two flow tubes, $\Delta O_x$, divided by the mean residence time ($\tau$) of
air inside the tubes:
$$P(O_x) = \frac{\Delta O_x}{\tau} = \frac{O_{x_{amb}} - O_{x_{ref}}}{\tau} \qquad\qquad (4)$$
A detailed schematic of the OPR instrument is shown in Figure 1. The two flow tubes exhibit the
same geometry and are made of quartz (14 cm-ID and 70 cm long). Each flow tube is connected to the
inlet and outlet flanges that are made of anodized aluminum and PTFE. Since a major issue previously
identified for the MOPS instrument was wall effects causing $NO_2$ losses (Cazorla and Brune, 2010),
the inner geometry of the flanges was designed based on fluid dynamics simulations using STAR
CCM+ V.8 (CD-adapco). The geometry was optimized to minimize radial mixing and recirculation
eddies that could increase wall effects. The design of the flanges can be found in the supplementary
material (Fig. S1).
Each flange consists of two parts. For both the inlet and outlet, a conical PTFE piece is screwed inside
an external aluminum flange. Four holes are drilled symmetrically around the aluminum flanges to
inject zero air around the PTFE inlet and to extract air around the PTFE outlet. The lengths of the inlet
and outlet flanges are 25 and 14 cm, respectively. The PTFE inlet has an external diameter of 2.54 cm
which increases to 7 cm over a length of 20 cm. The PTFE outlet starts from a diameter of 3 cm
which decreases to 1.27 cm over 10 cm. The aluminum flanges exhibit a curved conical inner surface
around the PTFE parts.
Ambient air is sampled through a common inlet (PFA, 1.27 cm-OD) at a flow rate of 4 L min$^{-1}$ and is
transferred into both flow tubes through the internal PTFE inlets (2 L min$^{-1}$), while additional zero air
(250 mL min$^{-1}$) is injected at the outer periphery of these inlets inside the flanges. This flow of zero
air helps keeping the ambient air flow forward, minimizing recirculation eddies, and therefore
reducing wall effects. The dilution of the sampled air is approximately 10%. At the outlet, air is
sampled only from the center of the flow tube, through the PTFE outlet (750 mL min$^{-1}$), while the rest





is extracted by an external pump (1.5 L min$^{-1}$). Both the injection and extraction of air are regulated
by mass flow controllers (MFC in Fig. 1).
The Ultem filter is placed on a rectangular aluminum frame outside of the reference flow tube, which
enables to flow ambient air between the filter and the flow tube using fans. This setup allows the two
flow tubes to be kept at the same temperature by extracting the heat released by the filter. For the
same reason, a frame covered by a FEP film (.002" thick, DuPont Teflon® FEP), transparent to the
solar radiation, is used for the ambient flow tube to reduce heat dissipation by the wind.
The air exiting the two flow tubes is mixed with 10 SCCM of NO (50 ppmv, Indiana Oxygen, USA),
leading to a NO mixing ratio of 650 ppbv in the conversion unit. The mixing of the gases takes place
in two identical pyrex chambers, providing a reaction time of approximately 22 sec at 20°C, which is
long enough to quantitatively titrate $O_3$ into $NO_2$. Both the relative humidity and temperature are
monitored in the air flow extracted from the flow tubes and at the $O_3$-to-$NO_2$ conversion unit.
Downstream the conversion unit, $O_x$ ($O_3$ + $NO_2$) is measured by an Aerodyne Cavity Attenuated
Phase Shift Spectroscopy (CAPS) $NO_2$ monitor (Kebabian et al., 2005;Kebabian et al., 2008). The
detection limit (3$\sigma$) for a 1-s integration time is 300 pptv. Since the CAPS is a single-cell monitor, the
measurements from the ambient and reference flow tubes are taken sequentially, using two solenoid
valves (SV1 and SV2 in Fig. 1). When air from the ambient (or reference) flow tube is sampled by the
CAPS monitor (750 ml min$^{-1}$), the same flow rate of air is extracted from the other flow tube by a
mass flow controller connected to a pump. The valves switch every 1 min, alternating the flows that
are sampled by the CAPS monitor and the pump. $\Delta O_x$ is calculated as the difference between an
ambient flow tube measurement and the average of 2 surrounding reference measurements, leading to
a $P(O_x)$ measurement every 2 min. The first 15 seconds of each 1-min measurement are discarded
since they describe a transient regime between ambient and reference flow tube measurements. Ozone
production values are calculated from Eq. (4).
The measurement sequence is automated and controlled through a National Instruments LabView
2013 interface. Three USB data acquisition boards are used (NI-9264, NI-6008, NI-6009) to control





the two solenoid valves and the seven mass flow controllers, as well as to record signals from the
CAPS monitor and sensors setup for humidity and temperature measurements.

**2.2    Laboratory and field experiments conducted to characterize the OPR**

Experiments conducted to characterize the OPR instrument include measurements of the mean
residence time, $O_x$ losses, and HONO production rates in the flow tubes and measurements of the $O_3$-
to-$NO_2$ conversion efficiency.
**The mean residence time** was quantified in each flow tube by injecting short pulses of toluene (10-s
in duration) at the inlet of the flow tubes. A PTR-ToFMS (Proton Transfer Reaction–Time of Flight
Mass Spectrometer, KORE Technology Inc.) was connected at the outlets to measure the time it takes
for a pulse introduced at the inlets to exit the flow tubes. The pulse experiment was repeated 5 times,
and the average was calculated as the mean residence time.
**$O_3$ and $NO_2$ losses** inside both flow tubes were measured in the laboratory and during the field
deployment described below by sampling mixtures of zero air and $O_3$ (or $NO_2$) at known mixing ratios
and by measuring $NO_2$ downstream the conversion unit (or directly at the exit of the flow tubes). A
relative loss was calculated from the difference in concentrations between the inlet and outlet and was
referenced to the inlet concentration. These tests were performed at relative humidity values ranging
from 0–65%.
The **release of HONO from the inner surface of the flow tubes** was quantified using a Chemical
Ionization Mass Spectrometer (CIMS, Georgia Tech). Mixtures of $NO_2$ and humid zero air were
introduced into the flow tubes, while HONO was measured both at the inlet and outlet. These
experiments were performed under dark conditions, as well as under various irradiated conditions
using artificial UV light provided by two types of fluorescent lamps: 4 lamps centered at 312 nm
(Vilber, T-15.M) and 4 lamps centered at 365 nm (Philips, T12).
Finally, the **$O_3$-to-$NO_2$ conversion efficiency** was measured by sampling zero air enriched with $O_3$
(3-170 ppbv) through the mixing chambers of the conversion unit, varying the flow of NO and
measuring $NO_2$ with the CAPS monitor. These tests were performed at various relative humidities





(25–60%). The conversion efficiency at a specific NO level was calculated from the ratio of $NO_2$
measured at this NO level to that measured when 700 ppbv of NO were added, assuming for the latter
that 100% of $O_3$ was converted. This assumption is verified from kinetic considerations
($k_{NO+O3}=1.80\times10^{-14}$ $cm^3$ $molecule^{-1}$ $s^{-1}$ and 23 s of residence time in the conversion unit) and from the
observation of a plateau for NO mixing ratios higher than 500 ppbv.
**2.3    Modeling experiments conducted to characterize the OPR**
As previously mentioned, the measurement principle of ozone production rates is based on the
assumption that (i) $P(O_x)$ in the ambient flow tube is similar to $P(O_x)$ in the atmosphere and (ii) there
is no significant production of ozone in the reference flow tube. Box model simulations were
performed to check whether this assumption is valid. In addition, simulations were also conducted to
investigate the impact on OPR measurements of (a) an $O_3$-to-$NO_2$ conversion efficiency lower than
100%, (b) $NO_2$ and $O_3$ losses and (c) HONO production inside the flow tubes, (d) a possible increase
of the temperature in the reference flow tube due to the UV filter, (e) the dilution of ambient air by
injecting zero air inside the flow tubes at the periphery of the inlets, and (f) reactions of OH with $NO_z$
species producing $O_x$.
**2.3.1    Selected data and chemical mechanism**
The simulations were performed using a box model based on the Regional Atmospheric Chemistry
Mechanism (RACM) (Stockwell et al., 1997). RACM is a gas-phase chemical mechanism developed
for the modeling of regional atmospheric chemistry and includes 17 stable inorganic species, 4
inorganic intermediates, 32 stable organic species and 24 organic intermediates for a total of 237
chemical reactions. Organic compounds are grouped together to form a manageable set of
compounds. Only 8 organic species are treated explicitly (methane, ethane, ethene, isoprene,
formaldehyde, glyoxal, methyl hydrogen peroxide and formic acid) and 24 are surrogates that are
grouped based on emission rates, chemical structure and reactivity with the OH radical.
Measurements from several field campaigns were used for this modeling exercise, including
measurements performed in (i) a megacity as part of the 2006 Mexico City Metropolitan Area
(MCMA-2006) (Dusanter et al., 2009b) and (ii) an urban area as part of the 2010 California Nexus





(CalNex) campaign (Griffith et al., 2016). Two days characterized by elevated and low $O_x$
concentrations were selected for each campaign and are presented in the supplementary material
(Table S1 and Fig. S2). For both campaigns, ozone was higher by approximately a factor 2 on high $O_3$
days ($\approx$ 100 ppbv) compared to low $O_3$ days ($\approx$ 50 ppbv). However, while both high and low ozone
levels were similar for the selected days of these campaigns, large differences were observed for $NO_x$
(6–120 ppbv) and OH reactivity (8–86 $s^{-1}$). Since OH reactivity and NOx are main drivers of ozone
production, these modeling results are expected to provide a good assessment of potential biases
associated to $P(O_x)$ measurement for any urban environments.
**2.3.2    Modeling of ambient $P(O_x)$ values**
The model was constrained by 10-min (MCMA) or 15-min (CalNex) average measurements of
temperature, pressure, humidity, organic and inorganic species, and J-values, while the differential
equation system was integrated by the FACSIMILE solver (MCPA Software Ltd). In total, 24 J-
values were used to constrain the model, as derived in Dusanter et al. (2009b), together with 7
inorganic and 17 organic species or surrogates. Tables reporting the constrained species and J-values
can be found in the supplementary material (Tables S2 and S3). The integration time was set at 30h
with constrained species reinitialized every two seconds. Ambient ozone production values were then
calculated from Eq. (1)–(3) and are referred as $P(O_x)_{atm}$ in the following. In total, 18 surrogates of
$RO_2$ species were taken into account to calculate $p(O_3)$ from Eq. (1), while 10 unsaturated surrogates
were used to calculate $l(O_3)$ from Eq. (2) (Table S4).
**2.3.3    Modeling of $P(O_x)$ values in the ambient and reference flow tubes**
Modeling OPR measurements requires simulating the chemistry inside each flow tube. J-values used
to model the chemistry in the ambient flow tube were the same as for the ambient modeling since the
quartz material used to build the flow tubes is transparent to solar irradiation. For the reference flow
tube, J-values were scaled based on the absorption coefficient of the Ultem film (Philipp et al., 1989)
as discussed in the supplementary material (section S2.1).
The model was constrained by the same meteorological parameters and chemical species as for
$P(O_x)_{atm}$. In addition, modeled concentrations of VOC-oxidation products and peroxy radicals



inferred from the modeling of $P(O_x)_{atm}$ were also constrained in these simulations (Table S5),
assuming that a significant fraction of the latters is not lost in the sampling line. The constrained
concentrations were initialized once, at the entrance of the flow tubes, and the simulations were run
for 10 minutes without reinitializing the constraints. The simulations were run separately for each
flow tube and P(O$_x$) was calculated every 15 s from Eq. (3). An integrated value of P(O$_x$) was then
computed for the flow tube residence time.
$P(O_x)_{atm}$ is compared to the integrated P(O$_x$) value from the ambient flow tube (referred as
$P(O_x)_{amb}$) to check whether ozone production in the ambient flow tube is similar to ambient ozone
production. The integrated value of P(O$_x$) in the reference flow tube (referred as $P(O_x)_{ref}$) is also
scrutinized to check whether ozone production is negligible in this flow tube.
**2.3.4    Modeling of OPR measurements**
Since the OPR instrument measures O$_x$ after conversion of O$_3$ into NO$_2$, NO$_2$ concentrations at the
exit of the conversion unit are calculated from the conversion efficiency C as shown in Eq. (5):
$$[NO_2]_{conv} = [NO_2]_\tau + C\,[O_3]_\tau \qquad\qquad : \qquad\qquad (5)$$
Here the concentrations reflect those observed at the exit of the conversion unit (subscript: *conv*) and
at the exit of the flow tubes (subscript: *τ)*. The concentrations at the exit of the flow tubes are the
model outputs at the residence time τ. Based on Eq. (4), the ozone production rate measured by the
OPR, P(O$_x$)$_{OPR}$, is then calculated from Eq. (6):
$$P(O_x)_{OPR} = \frac{[NO_2]_{conv,amb} - [NO_2]_{conv,ref}}{\tau} = \frac{[NO_2]_{\tau,amb} - [NO_2]_{\tau,ref} + C([O_3]_{\tau,amb} - [O_3]_{\tau,ref})}{\tau} \qquad (6)$$
In this equation the subscripts *amb* and *ref* indicate the ambient and the reference flow tubes,
respectively. A Bias in OPR measurements can be quantified by comparing $P(O_x)_{OPR}$ to $P(O_x)_{atm}$
assuming a conversion efficiency of 100% for the conversion units.
**2.3.5    Sensitivity tests**
The simulation performed without O$_x$ losses and HONO production in the flow tubes, no dilution, and
no temperature differences between the tubes will be referred as base simulation in the following. All



simulations performed including sensitivity tests are compared to the results from the base simulation
to assess the impact of operating conditions on ozone production measurements.
To assess the impact of a conversion efficiency lower than 100%, $P(O_x)_{OPR}$ is calculated from Eq. (6)
by varying the conversion efficiency using the model outputs from the base simulation. $P(O_x)$ values
inferred when varying the conversion efficiency are compared to values calculated for a conversion
efficiency of 100%. To account for $O_x$ losses, a similar sink of $O_3$ or $NO_2$ is introduced in the model
for each flow tube, with a first order loss rate ranging from $1.5\times10^{-4}$ to $1.2\times10^{-3}$ s$^{-1}$. This range of loss
rates corresponds to a relative loss of 4–28%. The measured $P(O_x)_{OPR}$ is again calculated by Eq. (6)
assuming a conversion efficiency of 100% and compared to the base simulation. Sensitivity tests were
also performed assuming that the loss of $NO_2$ on the quartz surface led to HONO formation with the
same first order rate as the $NO_2$ loss, or by including a HONO source in the model, independent of
$NO_2$, with production rates comparable to experimental observations. Additional sensitivity tests
focused on decreasing the constrained species by 5-30% to assess the impact of diluting ambient air in
the flow tubes, as well as increasing the temperature of the reference flow tube by 2% to 20% to
simulate a heat release by the UV filter. Finally, sensitivity tests were performed to investigate
whether reactions of OH with $NO_z$ species that produce $O_x$ could significantly impact the OPR
measurements. $NO_z$ species producing $NO_2$ or $NO_3$ ($NO_2$ reservoir) in the model when reacting with
OH are HONO, $HO_2NO_2$, organic nitrates, $HNO_3$, PANs and unsaturated PANs. The $NO_2$ and $NO_3$
products of the reactions mentioned above were removed from the model for the sensitivity test.
**2.4   Description of the field measurements**
The OPR instrument was deployed in the field, as part of the Indiana Radical, Reactivity and Ozone
Production Intercomparison (IRRONIC) campaign in Bloomington, Indiana, during July 2015. The
measurements were taken at the Indiana University Research and Teaching Preserve (IURTP) field
laboratory (39.1908N, 86.502W), 2.5 km northeast of the Indiana University Bloomington campus.
The site is a mixed deciduous forest containing northern red oaks and big-tooth aspens, which are
known to be strong emitters of isoprene and monoterpenes (Isebrands et al., 1999;Funk et al., 2005).
A highway (E Matlock Road, State Route 45) is located 1 km southwest, and therefore the site can be



impacted by anthropogenic emissions. The OPR flow tubes were setup on a scaffolding to expose
them to the sunlight for the entire day. The conversion units and the CAPS monitor were housed
inside the laboratory and were connected to the flow tubes using 4-m long heated ¼" PFA lines.
This campaign included measurements of OH, $HO_2^*$ ($HO_2+\alpha RO_2$), total peroxy radicals ($HO_2+RO_2$),
total OH reactivity, $NO_x$, $O_3$, anthropogenic and biogenic VOCs, radiation and meteorological data.
For the measurements presented in this publication, VOCs were measured by an online TD-GC/FID,
an online TD-GC/FID-MS (Badol et al., 2004;Roukos et al., 2009), and offline samplers for
DiNitroPhenylHydrazine (DNPH) cartridges (Waters Sep-Pak) and Sorbent cartridges (Carbopack
B/Carbopack C) by IMT Lille Douai. Measurements of NO (chemiluminescence, Thermo model 42i-
TL), $NO_2$ (cavity attenuated phase shift spectroscopy, Aerodyne Research), and ozone (2B Tech
model 202 sensor) were also conducted by the University of Massachusetts. Measurements of $J(NO_2)$
were performed using a scanning actinic flux spectroradiometer (SAFS, METCON) from the
University of Houston, while meteorological data, including temperature, relative humidity, wind
speed and wind direction were measured with a meteorological station from Montana State
University.
The OPR measurements were focused on investigating the sensitivity of P(Ox) to NOx (see section
3.3). This was achieved by introducing a certain amount of NO (ppbv range) inside the OPR sampling
line for 40 minutes, and then stopping the NO addition for another 40 minutes. This pattern was
repeated continuously, keeping the NO level constant for several days. The first 20 minutes of each
40-minutes measurements were discarded, since they correspond to a transient regime between the
disturbed-undisturbed $P(O_x)$ measurements due to the long air-exchange time in the flow tubes (see
section 3.1.1). The addition of NO in the OPR sampling line was performed through a 1/8"-OD
stainless steel tube using a NO cylinder (3.75 ppmv in $N_2$) from Indiana Oxygen and a mass flow
controller. After the mixing point, a length of 10 m of 1/2"-OD PFA tube was used as the sampling
line to ensure a good mixing of NO with the sampled air, leading to a residence time of approximately
10 s in the line at a total flow rate of 4 L $min^{-1}$.



## 3   Results and discussion

### 3.1   Laboratory characterization

#### 3.1.1   Quantification of the flow tubes residence time

As described in the experimental section, pulses of toluene were injected in the flow tubes to quantify the mean residence time. One of the 5 experiments that were conducted is shown in Figure 2. The pulse shape is asymmetric and exhibits a long tail, indicating that a large range of residence times is observed in the flow tubes. The toluene pulse is treated as a probability distribution of the time variable $t$, with the average residence time in the flow tubes being the mean of the probability distribution. The latter is calculated as a weighted average of the possible values that the time variable can take. The average residence time from the 5 toluene pulse experiments was $4.52 \pm 0.22$ min ($1\sigma$). The uncertainty reported for the residence time will lead to a 4.9% error ($1\sigma$) on the $P(O_x)$ measurements.

Based on the flow tube volume of 10.8 L and a total flow rate of 2.25 L min$^{-1}$ in each flow tube, a laminar plug flow would lead to a residence time of 4.79 min. The measured residence time is approximately 6% lower than the time calculated for laminar flow conditions, which is barely significant considering the $1\sigma$ uncertainty of 4.9% determined from the measurements. However, the asymmetry of the peak indicates that the flow rate at the central axis of the tube is larger, with the first molecules of toluene being sampled after approximately 2 minutes (Fig. 2). These observations are similar to that reported by Cazorla and Brune (2010) for sampling chambers exhibiting a different geometry and operated under different flow conditions. In this study, the authors reported a residence time that was 23% lower than plug flow calculations and a similar asymmetric shape for the pulse. Further work is needed to reduce the skewness of the time distribution.

Tests were also performed to quantify the air-exchange time in the flow tubes. These tests were performed by sampling a constant concentration of $O_x$ species with the OPR instrument until a stable $O_x$ signal was measured and by quickly changing the Ox concentration at the inlet. The time needed to reach 95% of a new stable $O_x$ signal was defined as the air-exchange time. The air-exchange time was quantified at approximately 20 minutes, corresponding to a maximum residence time of 1200 s. As

mentioned in section 2.1, a $P(O_x)$ value is recorded every 2 minutes. Since the air-exchange time is 20
minutes, the 2-minute $P(O_x)$ values are not independent from each other and therefore the OPR
instrument cannot detect rapid changes of $P(O_x)$. In order to get independent measurements of $P(O_x)$,
the OPR measurements are therefore averaged over 20 minutes.
**3.1.2    Quantification of $O_x$ losses in the flow tubes**
The principle of the OPR instrument requires that the only difference between the two flow tubes is
the suppression of gas-phase photolytic reactions leading to the formation of free radicals in the
reference tube. All other characteristics, including flow pattern and potential gas-wall interactions
should be the same in the two flow tubes so that they cancel out in the differential $O_x$ measurement.
However, if $O_x$ losses were slightly different between the two flow tubes, it could significantly impact
the $P(O_x)$ measurements. For example, a 2% difference in $O_x$ losses between the flow tubes would
lead to a bias of 27 ppbv $h^{-1}$ on the measurements for an ambient $O_x$ level of 100 ppbv and a residence
time of 4.5 min.
Figure 3 shows the results of $NO_2$ and $O_3$ loss tests for the two flow tubes, performed at different
dates during one month of field operation during the IRRONIC campaign and at different relative
humidity values. All $NO_2$ loss tests were performed under dark conditions, i.e. with both flow tubes
covered by an opaque cover. Figure 3-(a, c, e) shows that the $NO_2$ loss is lower than 5% in both flow
tubes and is close to 3% on average. When the two flow tubes are operated under the same conditions,
the relative loss in the reference tube seems to be higher than the loss in the ambient tube by only 1%
at most (Fig. 3-e). For an ambient $NO_2$ mixing ratio of 30 ppbv, a difference of 1% in $NO_2$ losses
between the flow tubes would lead to a 4 ppbv $h^{-1}$ bias in the $P(O_x)$ measurements.
Cazorla and Brune (2010) reported an uncertainty of ±14% for the MOPS instrument due to potential
differences in relative humidity between the two sampling chambers, which in turn leads to different
$NO_2$ losses. This was mainly due to a higher temperature in the reference chamber, which is covered
by the UV filter. However, the fans used in the OPR instrument to flow ambient air between the UV
filter and the flow tube minimize the temperature differences between the two tubes, leading to
relative humidity differences lower than 4%, as observed during the field testing. Figure 3-e also





shows that a decrease in relative humidity from 65% to 0% only leads to a small decrease of the $NO_2$
loss by 1–2%. A small difference of 4% in relative humidity between the two flow tubes is therefore
not expected to lead to additional errors in the $P(O_x)$ measurements. Further analysis of the impact of
$NO_2$ losses on the $P(O_x)$ measurements is discussed in the modeling results section.
Ozone loss tests were mainly performed under dark conditions. On 28 July however, $O_3$ losses were
measured with (a) the ambient flow tube exposed to the sunlight and the reference tube covered by the
UV filter (orange squares), (b) both flow tubes exposed to the sunlight (orange triangles) and (c) both
tubes covered by a dark cover (orange circles). For the first days of the campaign (29 June-8 July), a
close inspection of the measurement scatter shown in Figure 3-(b, d) indicates that the relative loss of
$O_3$ is at most close to 5%. However, ozone loss tests performed on 28 July, after one month of
operation in the field, reveal an increase of the relative loss up to 13-15%. Additional tests made in
the laboratory after the field deployments have shown that this loss can be reduced below 5% if the
quartz flow tubes are conditioned with elevated $O_3$ mixing ratios at high relative humidity for a few
days. These results indicate that unsaturated organic species may adsorb on the quartz surface and
may react with $O_3$.
Particular attention should be paid to the three different tests performed on 28 July regarding the
irradiation conditions. When the losses are quantified under dark conditions (orange circles in Fig. 3-
f), the losses are equal between the two flow tubes and close to 13%. However, when the ambient
flow tube is irradiated and the reference is covered by the UV filter (orange squares), it can be seen
that the relative loss in the ambient tube is higher than in the reference by approximately 3%. Box
modeling has shown that the gas-phase photolysis of $O_3$ in the ambient flow tube could at most
account for 0.05% of this additional ozone loss. Therefore, there seems to be a photo-enhanced ozone
loss that takes place when the ambient flow tube is irradiated. For an ambient $O_3$ level of 50 ppbv, this
difference in $O_3$ losses would lead to a negative $P(O_x)$ bias of approximately 20 ppbv $h^{-1}$. This photo-
enhanced loss of ozone is further discussed in the field deployment section (3.3).



### 3.1.3 Heterogeneous HONO production in the flow tubes

The formation of HONO in the flow tubes was investigated in the laboratory by sampling humid zero air (25-80% RH) enriched with $NO_2$ at various mixing ratios (0-100 ppbv) and by measuring HONO mixing ratios in the tubes as described above in section 2.2. Both clean and contaminated (used for more than one month during the IRRONIC campaign) flow tubes were tested to assess the magnitude of HONO production rates and to examine whether there is a dependence on $NO_2$ mixing ratios, humidity and irradiation. Mixing ratios of HONO up to 250 and 700 pptv were measured under dark conditions for clean and contaminated flow tubes, respectively. Higher mixing ratios of up to 1.5 ppbv were measured under irradiated conditions in the ambient flow tube ($J(NO_2)=1.4\times10^{-3}$ $s^{-1}$; $J(HONO)=3.1\times10^{-4}$ $s^{-1}$).

Dividing the measured mixing ratios of HONO by the residence time in the flow tubes (i.e. 4.5 min), an average production rate can be calculated under dark and irradiated conditions. It is important to note, however, that HONO is also photolyzed at the wavelengths emitted by the lamps (312 nm and 365 nm) and production rates calculated under irradiated conditions represent lower bounds. It is estimated that for the J(HONO) value mentioned above and a negligible loss of HONO from OH+HONO, the HONO production rate will be underestimated by less than 8%. The dark HONO production is on the order of 9 ppbv $h^{-1}$ in both flow tubes, while the total HONO production under irradiated conditions (dark + photo-enhanced) can reach up to 20 ppbv $h^{-1}$ in the ambient flow tube. In the reference flow tube, the UV light did not impact the formation of HONO, since wavelengths below 400 nm are blocked by the UV filter.

The HONO production rate was not observed to depend on $NO_2$ or humidity and HONO could be even released when no $NO_2$ was introduced into the contaminated flow tubes. These results strongly suggest that nitro-containing compounds and organic photosensitizers were adsorbed on the walls of the flow tubes and that the HONO production rate depends on contamination levels. Indeed, it was observed that flowing humid zero air in the flow tubes for a few days could reduce the HONO production rate to negligible levels.



### 3.1.4 Quantification of the conversion efficiency

Based on kinetic considerations for the titration reaction of $O_3$ by NO, i.e. a rate constant of $1.80 \times 10^{-14}$ $cm^3$ $molecule^{-1}$ $s^{-1}$ at 298K (Atkinson et al., 2004), a reaction time of 23 seconds, and the addition of 500 ppbv of NO in the conversion unit, an $O_3$-to-$NO_2$ conversion efficiency of 99.5% is expected. These calculations are shown in Figure 4 (black solid line) for different mixing ratios of NO (50–800 ppbv) together with laboratory measurements (symbols) made at different $O_3$ levels. This figure shows that a plateau of almost 100% of conversion is observed at NO mixing ratios higher than 500 ppbv. These experimental results are in good agreement with the calculated curve, although the measurements performed at a low $O_3$ mixing ratio of 3.5 ppbv slightly underpredict the curve for NO mixing ratios lower than 500 ppbv. However, the conversion plateau is reached for all $O_x$ levels and both conversion units (one for each flow tube) for NO mixing ratios higher than 500 ppbv. During the field deployment of the instrument, an NO mixing ratio of 650 ppbv was used to ensure that the difference in conversion efficiency between the two mixing chambers was lower than 0.1% and could be assumed to be 100% for both chambers.

In the first version of MOPS (Cazorla and Brune, 2010) the $NO_2$-to-$O_3$ conversion was performed by photolyzing $NO_2$ using a light-emitting diode, achieving a maximum conversion efficiency of 88% at 17 ppbv of $NO_2$. In the most recent version of the instrument (Baier et al., 2015), the conversion efficiency was increased to 88–97% for $NO_2$ mixing ratios lower than 35 ppbv using a highly-efficient UV lamp that provided ten times more photons than the light-emitting diodes. In the MOPS instrument, however, the conversion efficiency depends on $NO_2$ levels, as well as on the intensity of the lamp that could drift during a long period of use in the field. In the OPR instrument, the conversion efficiency is stable and does not depend on $O_3$ mixing ratios. On the other hand, an NO cylinder is required to perform the conversion and possible $NO_2$ impurities in the cylinder have to be monitored. Indeed, $NO_2$ impurities coming either from the NO mixture or from NO oxidation in the lines were observed, but were kept at low levels of approximately 6–10 ppbv. Since this impurity is present in both the ambient and reference channel, it does not affect the $P(O_x)$ determination.





### 3.1.5 Detection limit of the OPR

The detection limit (DL) of the CAPS monitor was quantified by sampling zero air for several hours after several days of conditioning with ambient air. The time resolution was set to 1 s and the zero measurements were averaged over 45 s segments, corresponding to the OPR measurement averaging time. The detection limit ($3\sigma$) for a 45 s integration time was quantified at 34 pptv. This detection limit for $NO_2$ together with a residence time of 4.5 min in the flow tubes should lead to a detection limit of 0.6 ppbv $h^{-1}$ for 2-min $P(O_x)$ measurements (1-min measurement from each flow tube). However, nighttime measurements made during the IRRONIC field campaign revealed that the measurement scattering for the complete setup (flow tubes + $O_3$-to-$NO_2$ conversion unit + CAPS) was significantly larger than that expected from the noise of the CAPS monitor. Based on the observed nighttime $1\sigma$ variability of 2.1 ppbv $h^{-1}$, a limit of detection ($3\sigma$) of 6.2 ppbv $h^{-1}$ was inferred for the OPR instrument. The scatter in $P(O_x)$ measurements does not only depend on the precision of the CAPS monitor, but also depends on how fast each flow tube responds to variations of $O_x$ at the inlet. Indeed, if the time constant for the response is slightly different between the 2 flow tubes, fluctuations of $O_x$ species at the inlet will introduce some scatter in the OPR measurements. In addition, small changes in temperature and humidity may evenly affect $O_x$ losses in each flow tube, leading to additional scatter in the $P(O_x)$ measurements.

### 3.2 Numerical Modeling

As mentioned in the experimental section, several days from different field campaigns were selected to model ambient $P(O_x)$, $P(O_x)$ in both flow tubes, and the impact of some operating conditions on the OPR measurements. The results from 30 May 2010 of the CalNex field campaign were selected to illustrate the discussion and results from the other days are shown in the supplementary material (Figs. S4, S5, S7-S9). A detailed analysis of the chemistry occurring in each flow tube is discussed below to assess the reliability of OPR measurements.

### 3.2.1 Radical budget in flow tubes

An analysis of the radical budget was performed in each flow tube to gain insights into the processes driving radical production and loss routes. Figure 5 shows the production and loss rates of OH (upper





panel) and peroxy radicals (lower panel) for each flow tube on 30 May 2010 during CalNex. The
production and loss rates were calculated taking into account initiation, propagation and termination
processes as described below.
OH production rates were calculated from photolytic reactions involving closed shell molecules ($O_3$,
HONO, $H_2O_2$, $HNO_3$, $HO_2NO_2$ and organic peroxides), reactions of $O_3$ with alkenes, and the
propagation of $HO_2$ by reaction with NO. Loss routes of OH includes propagation reactions to $HO_2$
and $RO_2$ by reaction with CO and VOCs and termination reactions of OH with $NO_2$ and other species
(NO, PANs, $HNO_3$, HONO and $HNO_4$). For peroxy radicals, production routes include the photolysis
of organic species (carbonyls, organic peroxides and organic nitrates), the ozonolysis of alkenes, PAN
decomposition, and the propagation of OH. Loss routes were calculated from reactions of peroxy
radicals with $NO_x$, self or cross reactions between peroxy radicals and propagation of $HO_2$ to OH.
Figure 5 clearly shows that the UV filter covering the reference flow tube leads to a decrease of the
initiation rates of all radicals by more than a factor of 10 and a decrease of their propagation rates by
at least a factor of 30. In the ambient flow tube, photolytic reactions of OVOCs are the most important
initiation routes of peroxy radicals, with a contribution of approximately 95%. HONO and $O_3$
photolysis are the most important initiation routes of OH, contributing by approximately 45% each. In
the reference flow tube, the primary route of radical initiation is $O_3$-alkenes reactions since
wavelengths below 400nm are suppressed.
The propagation reactions are important in both flow tubes for the production and loss of OH and
peroxy radicals. However, the partitioning between initiation and propagation processes is different in
the two tubes, which in turn leads to different OH chain lengths. The OH chain length is calculated as
the rate of propagation of $HO_2$ to OH divided by the total initiation of $RO_x$ radicals. As can be seen
from Figure 5, the OH chain length is fairly constant at a value of 3 in the ambient flow tube, while in
the reference flow tube it quickly decreases to unity for most of the day and to values lower than 1 in
the late afternoon. Therefore, in addition to lowering initiation rates of radicals, the UV filter allows to
reduce ozone production by lowering the cycling efficiency within the pool of $RO_x$ radicals.





A close inspection of the radical termination rates in Figure 5 indicates that the peroxy-$NO_x$
termination reactions are almost suppressed in the reference flow tube. This observation is also
supported by Figure S6, which shows time series of the peroxy radicals ($HO_2$ and $RO_2$) and NO in
each flow tube at a residence time of 4.5 min. Since $NO_2$ photolysis is almost eliminated in this tube,
the $O_3$-$NO_x$ PSS is shifted towards $NO_2$ due to the reaction of NO with $O_3$. As a result, NO mixing
ratios in the reference flow tube are at least one order of magnitude lower than in the ambient flow
tube. The propagation rate from $HO_2$+NO is therefore reduced and the OH+$NO_2$ loss route is
enhanced, leading to the shorter OH chain length discussed above. It is also interesting to note that
peroxy radical mixing ratios in the reference flow tube are on the same order of magnitude as in the
ambient flow tube. This counterintuitive observation is also due to the consumption of NO in the
reference flow tube that leads to a longer lifetime for the peroxy radicals, as shown in Figure S6.
Calculating $P(O_x)$ from Equations (1-3) results in ozone production rates in the ambient flow tube,
$P(O_x)_{amb}$, in good agreement with the modeled $P(O_x)_{atm}$ values, as shown in Figure 6, with a small
underestimation of approximately 10% on average. However, significant ozone production rates are
also observed in the reference flow tube, which can reach up to 4 ppbv h$^{-1}$ on this day, while higher
values were observed on other days (e.g. 30 ppbv h$^{-1}$ on 21 March 2006 of the MCMA-2006
campaign, Figure S10 in the supplementary material). Ozone production rates in the reference flow
tube are about 10–15% of that observed in the ambient flow tube for most of the day. It is important to
note, however, that this ozone production is in reality $O_x$ (=$O_3$+$NO_2$) production, since $NO_2$ photolysis
is almost suppressed in the reference flow tube. These results indicate that the assumptions initially
made on the principle for $P(O_x)$ measurements, i.e that $P(O_x)$ in the ambient flow tube mimics $P(O_x)$
in the atmosphere and $P(O_x)$ in the reference flow tube is not significant, are not completely fulfilled.
Based on the modeling results discussed above, the accuracy of the measurements could be
significantly impacted by $O_x$ production in the reference flow tube.
$P(O_x)_{OPR}$ was calculated from Eq. (6), using an $O_3$-to-$NO_2$ conversion efficiency of 100%, and is
also shown in Figure 6. As discussed above, $P(O_x)_{OPR}$ underestimates the modeled $P(O_x)_{atm}$,
mainly due to significant $O_x$ production in the reference flow tube. The scatter plot shown as insert in



this figure indicates that a negative bias of approximately 20% would be observed for $P(O_x)$
measurements performed on this day. A negative bias ranging from 15–20% was observed during the
other three days that were modeled (Figure S11).
As mentioned in the experimental section, concentrations of peroxy radicals obtained as model
outputs from the modeling of $P(O_x)_{atm}$ were constrained for the simulations inside the flow tubes,
assuming that most of these species are not lost if a short high-flow rate sampling inlet is used.
However, simulations were also performed without constraining the peroxy radicals to assess the
impact on the simulation results. These simulations have shown that $P(O_x)$ are lower by 10% and 30%
in the ambient and reference flow tubes, respectively, when peroxy radicals are not constrained.
Overall, the measured ozone production, which is the difference between $P(O_x)$ in the two flow tubes,
would only decrease by 2-4%. Therefore, not constraining peroxy radicals in the simulations does not
impact the comparison between $P(O_x)_{atm}$ and $P(O_x)_{OPR}$, with $P(O_x)_{OPR}$ underestimating $P(O_x)_{atm}$
by 15-20 %.
However, the reason for this disagreement depends on whether peroxy radicals are constrained. When
peroxy radicals are constrained, the disagreement is mainly caused by $O_x$ production in the reference
flow tube. On the opposite, when peroxy radicals are not constrained, this disagreement is due to an
underestimation of $P(O_x)_{atm}$ by $P(O_x)_{amb}$. This underestimation is the result of a latency in the first
part of the ambient flow tube due to the time needed to reproduce the radicals, which is on the order
of 1-2 minutes. It is very likely that only a fraction of the peroxy radicals will be transferred to the
flow tubes and a combination of the two issues discussed above will lead to the negative bias of 15-

21    20%.

**3.2.2   Sensitivity tests - Assessment of the impact of operating conditions on OPR**

23           **measurements**

Figure 7 shows the dependence of $P(O_x)_{OPR}$ on the $O_3$-to-$NO_2$ conversion efficiency, $O_3$ and $NO_2$
surface-losses, surface-production of HONO, and a dilution of the sampled air. The results are
displayed for two different times of the day, characterized by different $O_3$ and $NO_2$ mixing ratios,
which have been identified as upper (orange squares) and lower (blue squares) limits for the impact





on the P(O$_x$) measurements. In addition, these results are also displayed using daily averaged values
(green triangles), which are more representative of the average impact of a particular parameter on
P(O$_x$) measurements. The figures described below are for the CalNex campaign during 30 May 2010.
Results from the other days are shown in the supplementary material (Figures S12-S14).
Figure 7-a shows that $P(O_x)_{OPR}$ is very sensitive to the O$_3$-to-NO$_2$ conversion efficiency. For
instance, a conversion efficiency of 85% would lead to an underestimation of the P(O$_x$) measurements
by 20–60% (≈35% on average), depending on the chemical composition of the air mass. It is
interesting to see that the change in  $P(O_x)_{OPR}$, expressed as the ratio between $P(O_x)_{OPR}$ at a
conversion efficiency lower than 100% and $P(O_x)_{OPR}$ at a conversion efficiency of 100% (base
simulation), changes linearly with the conversion efficiency. The slope of the straight line can be used
as an indicator to gauge the impact of the conversion efficiency on P(O$_x$) measurements throughout
the day. As can be seen from Equation (6), for the limiting case of C=0, the measured P(O$_x$) is
determined by the absolute NO$_2$ difference between the two flow tubes. The O$_3$–NO$_x$ PSS is shifted
towards NO$_2$ in the reference flow tube, due to the lack of NO$_2$ photolysis, reducing the NO$_2$
difference between the two tubes and lowering the measured P(O$_x$). These results stress out the need
to reach a conversion efficiency better than 98% to keep this artifact below 5%. The OPR instrument
described in this study exhibits a conversion efficiency higher than 99.9% and is not impacted by this
issue.
Relative surface-losses of 3% and 5% have been observed for NO$_2$ and O$_3$, respectively, during the
laboratory and field testing (section 3.1.2). Figure 7-b shows that a relative NO$_2$ loss of 3% in the flow
tubes can lead to an overestimation of up to 8% (≈3% on average). On the other hand, Figure 7-c
shows that a 5% relative loss of O$_3$ can lead to an underestimation of up to 30% (≈5% on average).
These contrasting effects can be explained as follows; ozone in the reference flow tube is lower than
in the ambient flow tube, due to the conjunction of a lower production rate of ozone and a shift of the
O$_3$–NO$_x$ PSS towards NO$_2$. A similar relative loss of ozone in the two flow tubes will therefore lead to
a larger absolute loss of O$_x$ species in the ambient flow tube, which in turn will lead to an
underestimation of the P(O$_x$) measurements (Eq. (6)). In contrast, NO$_2$ is higher in the reference flow





tube and a loss of $NO_2$ will lead to a larger absolute loss of $O_x$ species in the reference flow tube, and
as a consequence, to an overestimation of the $P(O_x)$ measurements.
Figure 7-d shows how an heterogeneous production of HONO can impact the $P(O_x)$ measurements. In
these simulations, a HONO source was added in the model, with a production rate of 10 ppbv $h^{-1}$ in
both flow tubes (dark HONO production) and an additional varying production rate in the ambient
flow tube (enhanced HONO production). The x-axis presents the HONO production rate in the
ambient flow tube, where 10 ppbv/h corresponds to the dark production only. Moreover, this figure
indicates that HONO production rates of 20 ppbv $h^{-1}$ in the ambient flow tube, similar to experimental
observations, can lead to an overestimation of the $P(O_x)$ measurements by up to 40% (≈27% on
average). This overestimation results from HONO photolysis in the ambient tube, which leads to
additional OH production, which in turn leads to an enhancement of VOC oxidation rates and ozone
production. Additional simulations were also performed assuming that $NO_2$ molecules lost on the
surface were equally converted into HONO in both flow tubes (Fig. 7-f), although it is unlikely that
the conversion yield of $NO_2$ into HONO is 100%. The results indicate that, for a relative $NO_2$ loss of
3%, $P(O_x)$ could be overestimated by up to 15% (10% on average). Note that the impact of this
HONO formation adds up to the previously discussed overestimation due to the $NO_2$ loss.
Figure 7-e displays how the injection of zero air at the periphery on the PTFE inlets impacts $P(O_x)$
measurement through a dilution of the sampled air. As can be seen from this figure, a 10% dilution
leads to less than 9% underestimation of $P(O_x)$.
Additional sensitivity tests (not shown) were performed to test the impact of a temperature increase in
the reference flow tube due to heat release by the UV filter, as well as reactions of OH with $NO_z$
species that produce $NO_2$. A temperature increase of 5% in the reference flow tube ($1^o$ C increase at
$20^o$ C) can lead to an underestimation of up to 5%, while the $O_x$ production from reactions of OH with
$NO_z$ species can lead to an overestimation of up to 3%.



### 3.2.3   Conclusions on potential biases on $P(O_x)_{OPR}$ measurements

From the above discussion, we can conclude that there are two main sources of errors. The first source of errors is due to $O_x$ production in the reference flow tube and the latency for $RO_x$ reformation in the ambient flow tube, the extent of each depending on the fraction of ambient peroxy radicals that is transmitted into the flow tubes. The combination of these two issues can lead to an underestimation of ambient $P(O_x)$ by 15-20% on average for the conditions observed during MCMA-2006 and CalNex-2010. The second main source of errors is caused by a surface-production of HONO in the ambient flow tube. Based on a HONO production rate of 20 ppbv $h^{-1}$, $P(O_x)$ would be overestimated by approximately 30% on average. Additional sources of errors are due to the 4.9% uncertainty on the flow tube residence time, 5% $O_3$ and 3% $NO_2$ surface-losses, the dilution by 10% of the sampled air, a possible temperature increase of 5% in the reference flow tube and $O_x$ production from reactions of OH with $NO_z$ species. Daily averaged values and upper bounds of errors associated with these factors, as derived from all modeled days, are reported in Table 1.

Based on the daily average values reported in Table 1, direct sums of the potential negative and positive biases lead to -44% and +40%, respectively. However, the magnitude of each error will depend on atmospheric composition and positive errors will, to some extent, cancel out with negative errors. A quadratic sum of all these potential errors leads a range of ±36%. The estimation of these errors are based on ambient conditions observed in two different environments, with different air compositions for 4 different days. It is safe to assume that similar error values would be observed in other urban environments.

### 3.3   Current limitations for field operation

As mentioned in section 2.4, OPR measurements were performed during the IRRONIC field campaign. Figure 8 displays time series for a subset of measurements performed from 10-14 July 2015, including two anthropogenic VOCs (toluene and acetylene), a biogenic VOC (isoprene) and inorganic species ($O_3$, NO and $NO_2$). It is clear from this figure that the measurement site was mainly impacted by biogenic emissions, with isoprene reaching at least 5 ppbv most of the days, while anthropogenic VOCs were low (<500 pptv). In addition, $NO_x$ levels were lower than 3 ppbv on these





days, confirming the low impact of anthropogenic emissions. These observations indicate that the
photochemistry was mainly driven by the oxidation of biogenic VOCs under low $NO_x$ conditions,
similar to that observed in other forested areas (Griffith et al., 2013). Isoprene is very reactive with the
hydroxyl radical and the strong diurnal variation of this species led to a large range of OH reactivity
(a few $s^{-1}$ up to 30 $s^{-1}$, not shown). The conjunction of the latter with low levels of $NO_x$ makes this site
of particular interest to study the sensitivity of ozone formation to $NO_x$ by adding $NO_x$ in the OPR
instrument as described in the experimental section (section 2.4).
Due to the low levels of ambient NOx, ozone production rates at the site were lower than the OPR
detection limit of 6.2 ppbv $h^{-1}$ (section 3.1.5). Indeed, $P(O_x)$ calculations based on total peroxy radical
measurements performed using the Peroxy Radical Chemical Amplifier technique indicated peak
ozone production rates of approximately 2 ppbv $h^{-1}$ (not shown). Ambient measurements performed
by the OPR instrument without addition of NO should therefore be scattered around zero within the
measurement precision. Figure 8 also displays $\Delta O_x$ values (difference in $O_x$ mixing ratios between the
two flow tubes) measured by the instrument without the addition of NO ($\Delta O_x^{zero}$, blue diamonds).
While $\Delta O_x^{zero}$ was scattered around zero during nighttime, it consistently exhibited large negative
values during daytime (–1 to –5 ppbv), indicating that $O_x$ mixing ratios in the ambient flow tube were
lower than in the reference flow tube.
It is interesting to note that $\Delta O_x^{zero}$ values are anticorrelated with $J(NO_2)$ (Fig. 8). Covering the
ambient flow tube with a similar UV filter than the reference flow tube, i.e. operating the two tubes
under similar irradiation, showed that $\Delta O_x$ increases towards less negative values and ultimately
reaches zero. This behavior indicates that the higher loss rate of Ox species in the ambient flow tube is
due to the solar irradiation and points towards a photo-enhanced surface loss of $O_x$ species initiated by
photons at wavelengths lower than 400 nm. As ambient $NO_2$ mixing ratios were much lower than the
observed loss of $O_x$, this photo-enhanced loss involves a loss of ozone. For an ambient $O_3$ level of 40
ppbv, as usually observed during the field measurements, a $\Delta O_x^{zero}$ of -3 ppbv corresponds to a 7.5%
difference in $O_3$ losses between the two flow tubes and an ozone loss rate higher by approximately 39
ppbv $h^{-1}$ in the ambient flow tube compared to the reference flow tube. This issue was further



investigated in the laboratory. Tests performed using artificial irradiation and mixtures of humid air
and ozone confirmed that light-induced processes at wavelength lower than 400 nm lead to a loss of
ozone at the surface of the ambient flow tube. It was found that this loss depends on ambient ozone
levels, J-values and absolute humidity.
This version of the OPR instrument is therefore not suitable to perform ambient $P(O_x)$ measurements
since the measured $\Delta O_x$ is a combination of ambient ozone production and surface-$O_3$ losses in the
ambient flow tube. For this reason, the OPR measurements were focused on investigating the
sensitivity of $P(O_x)$ to $NO_x$, by recording the relative change in $P(O_x)$ when the chemical composition
of ambient air was perturbed by an addition of NO. For these measurements, it is assumed that $\Delta O_x^{zero}$
is representative of the instrumental zero and $\Delta O_x^{zero}$ measurements are referred as "baseline" in the
following. $\Delta O_x$ measurements performed with an addition of NO are assumed to deviate from $\Delta O_x^{zero}$
due to a change in ozone production in the ambient flow tube, while the surface loss of ozone is
assumed to be unchanged. This measurement step is denoted $\Delta O_x^{NO}$. The difference between $\Delta O_x^{zero}$
and $\Delta O_x^{NO}$ divided by the residence time in the flow tubes therefore provides a quantification of the
change in $P(O_x)$, refered as $\Delta P(O_x)$, due to the addition of NO. The validity of the assumption that the
$O_3$ photo-enhanced surface-loss is not disturbed by the addition of NO is discussed below.
Investigating the ozone production sensitivity to NO is outside the scope of this paper and we only
present measurements performed when 6 ppbv of NO were added in the instrument to illustrate its
current performances and limitations. Figure 8 displays time series of $\Delta O_x^{NO}$ (orange diamonds) when
6 ppbv of NO were added in the flow tubes. When NO is added, there is almost no change in $\Delta O_x$
during nighttime. In the absence of sunlight, NO only converts $O_3$ into $NO_2$ and the amount of $O_x$
measured by the CAPS monitor does not change. During daytime, $\Delta O_x^{NO}$ is higher than $\Delta O_x^{zero}$,
suggesting production of ozone in the ambient flow tube. The difference between $\Delta O_x^{NO}$ and $\Delta O_x^{zero}$,
divided by the residence time in the flow tubes, represents the change in ozone production rates and is
displayed in the bottom panel of figure 8 as $\Delta P(O_x)$. Changes in ozone production of up to 20 ppbv h$^-$
$^1$, well correlated with $J(NO_2)$, are observed for these days. Ozone production being $NO_x$–limited in



this environment, a positive change in $P(O_x)$ is indeed expected when a small amount of $NO_x$ is added
to the flow tubes.
However, the assumption that the photo-enhanced surface-loss of ozone does not change when NO is
added may breakdown for large NO mixing ratios. Indeed, the addition of NO in the flow tubes leads
to the conversion of a significant fraction of $O_3$ into $NO_2$, which in turn reduces the absolute loss of
$O_3$ in the ambient flow tube, leading to a shift of the $\Delta O_x^{zero}$ baseline to less negative values. $\Delta P(O_x)$
values reported in Figure 8 will therefore be the combination of a change in ozone production and a
change in the absolute loss of $O_3$. If the change in the ozone loss rate is significant compared to the
change in the ozone production rate, this could lead to an overestimation of the change in ozone
production. An assessment of this measurement bias requires modeling the chemistry in both flow
tubes to separate the two contributions, i.e the changes in (i) ozone production and in (ii) ozone loss.
While this work is outside the scope of this publication, which focuses on the performances and
limitations of the OPR instrument, it is interesting to note that preliminary modeling indicates a bias
lower than 5 pbbv h$^{-1}$ when 6 ppbv of NO is added.
The field deployment during IRRONIC revealed an additional bias in $P(Ox)$ measurements due to a
photo-enhanced loss of ozone at the inner surface of the ambient flow tube and the difficulty to probe
changes in $P(O_x)$ when the sampled air mass is perturbed by an addition of NO. Ambient
measurements of $P(O_x)$ with the current version of the OPR would necessitates performing frequent
zeros of the instrument to track the ozone loss and unfortunately a simple solution to do so was not
found. This work shows that the sampling part of the OPR instrument needs to be rethought to remove
(or reduce to a negligible level) the photo-enhanced surface-loss of ozone, which is a prerequisite to
get an instrument capable of reliable measurements of ozone production rates.
**3.4   Comparison to the MOPS instrument and potential improvements for the OPR**
**instrument**
Previous studies (Cazorla and Brune, 2010;Baier et al., 2015) have shown that measurements of
ambient ozone production rates are feasible. Baier et al. (2015) reported that the zero of their MOPS
instrument was achieved by removing the UV filter from the reference chamber for a full day to



record a diurnal profile of $\Delta O_x$, which was then subtracted from the raw $\Delta O_x$ measurements on other
days. This zeroing procedure was also tested on the OPR instrument, but led to unrealistically high
ambient $P(O_x)$ values of approximately 40 ppbv h$^{-1}$ for the low-NO$_x$ forested environment of
IRRONIC. This result also suggests that altering the irradiation conditions of the OPR flow tubes
leads to a wrong zero of the instrument. This zeroing technique seems to provide better results for the
MOPS instrument and it is possible that the design used for the MOPS sampling chambers or the
material used to build them (FEP) make it less sensitive to light-dependent surface reactions.
Since the main artifacts on the OPR instrument are caused by heterogeneous surface-reactions in the
flow tubes, i.e. HONO production (section 3.2.2) and ozone losses (section 3.2.2 and 3.3), the flow
tubes should be redesigned to reduce the impact of physicochemical processes occurring near the
quartz surface on the ozone production chemistry occurring at the center of the tubes. For instance,
the diameter of the tubes could be increased to reduce the surface-to-volume ratio, and their lengths
could be shortened together with an increase of the total flow rate to reduce the contact-time between
trace gases and the walls. A shorter residence time would also lead to a shorter air-exchange-time,
which in turn would help minimizing the scatter in $\Delta O_x$ measurements and would help improve the
time resolution necessary to generate independent $P(O_x)$ measurements. However, a shorter residence
time would also lead to a lower detection limit and a tradeoff between these 2 parameters will likely
have to be made.
Another solution worth investigating would be to minimize surface reactions by coating the inner
surface of the flow tubes with Teflon or by applying a chemical treatment on the quartz surface,
which should help removing reactive sites. The latter has already been applied for laboratory kinetic
experiments to clean reactor surfaces. Interestingly, it was reported that this type of treatment can also
reduce HONO production on quartz surfaces (Laufs and Kleffmann, 2016).
Regarding the deployment of this OPR instruments in the field, a reliable zeroing method would be
suitable for both ambient $P(O_x)$ and $P(O_x)$ sensitivity measurements. An interesting solution would be
to introduce a radical scavenger in the flow tubes to supress ozone production, but a suitable
compound has yet to be identified.



**4    Conclusions**
An instrument dedicated to direct measurements of ozone production rates (OPR) was developed and
consists of two quartz flow tubes, an $O_3$-to-$NO_2$ conversion unit and an Aerodyne CAPS $NO_2$
monitor. This setup, compared to the $NO_2$-to-$O_3$ conversion approach previously published in the
literature, presents the advantage of a conversion efficiency higher than 99.9%, which is independent
of ambient $O_x$ levels. Laboratory and field testing performed to characterize the performance of this
instrument showed that dark losses of $O_3$ and $NO_2$ inside the flow tubes are lower than 5% and 3%,
respectively. However, it was shown that dark ozone losses can increase after a long exposure of the
flow tubes in the field and frequent cleaning steps should be performed during nighttime by flowing
humid air and $O_3$ in the tubes to keep the loss below 5%.
A modeling exercise taking advantage of measurements from previous urban field campaigns showed
that a latency in ozone production in the ambient flow tube and a net ozone production in the
reference flow tube can lead to a 18% measurement underestimation of ambient $P(O_x)$ on a daily
average for the conditions of the MCMA–2006 and CalNex–2010 field campaigns. However, the
magnitude of this underestimation depends on the chemical composition of ambient air and it is
recommended to assess this potential bias for future campaigns.
Sensitivity tests performed during the modeling exercise highlighted the importance of a high
conversion efficiency, since a conversion of 95%, which is only 5% lower than the maximum, could
lead to an underestimation of ambient $P(O_x)$ by approximately 20% on a daily average for the two
selected field campaigns. A dark surface loss of ozone in the flow tubes would lead to an
underestimation of ambient $P(O_x)$, while a $NO_2$ loss would lead to an overestimation. On a daily
average, an underestimation of 10% and an overestimation of 5% were assessed for an $O_3$ loss of 5%
and an $NO_2$ loss of 2%, respectively. A photo-enhanced production of HONO in the ambient flow
tube on the order of 20 ppbv $h^{-1}$ would also lead to an overestimation of ambient $P(O_x)$ by 27% on a
daily average. Overall, a quadratic sum of these potential biases for the conditions of the two urban
field campaigns leads to a range of errors of ±37% on a daily average.





As shown from the first deployment of the OPR instrument, there is an additional bias due to a photo-
enhanced loss of $O_3$ taking place in the ambient flow tube. This requires improving the sampling
design to be able to perform reliable ambient measurements. The first field deployment of the OPR
instrument was performed in a low $NO_x$ environment, allowing focusing the study on the sensitivity
of ozone production to $NO_x$. Significant changes in ozone production rates were observed (up to 20
ppbv h$^{-1}$) when 6 ppbv of $NO_x$ were added in the flow tubes, consistent with a $NO_x$-limited production
regime.
**Acknowledgements**
This work was supported by grants from the Regional Council Nord–Pas-de-Calais through the
MESFOZAT project, as well as the French National Research Agency (ANR–11–LABX–0005–01)
and the European Funds for Regional Economic Development (FEDER) through the CaPPA
(Chemical and Physical Properties of the Atmosphere) project. The authors thank the Région Hauts-
de-France and the Ministère de l'Enseignement Supérieur et de la Recherche (CPER Climibio) and
the European Fund for Regional Economic Development for their financial support. The authors are
grateful to Dr. William Bloss and Dr. Leigh Crilley (Birmingham University) for sharing their
experience on the OPR technique and for the idea of using quartz flow tubes as sampling chambers
for the OPR instrument. The authors are also grateful to Vinod Kumar and Vinayak Sinha (IISER
Mohali) who provided support and assistance during the initial development stage of the OPR
instrument. Finally, the authors thank the Mechanical Instrument Services at Indiana University for
the construction of the flow tube flanges.





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





**Table 1.** Sources of errors on $P(O_x)$ measurement. Upper limits and campaign averages of errors assessed from modeling the selected days of the MCMA-2006 and CalNex-2010 field campaigns (see text). FT: Flow Tube

| Sources of errors | Value | Negative bias on $P(O_x)$ | | Positive bias on $P(O_x)$ | |
|---|---|---|---|---|---|
| | | average | (upper limit) | average | (upper limit) |
| Residence time (s) | 271 ± 13[*] | -4.9%[*] | (-4.9%[*]) | +4.9%[*] | (+4.9%[*]) |
| $O_3$ production in ref. FT & latency in amb. FT | | -18%[**] | (-20%[**]) | | – |
| $O_3$ loss | 5%[*] | -10%[**] | (-25%[**]) | | – |
| $NO_2$ loss | <3%[*] | | – | 5%[**] | (+11%[**]) |
| HONO production | up to 20 ppbv/h[*] | | – | +27%[**] | (+40%[**]) |
| Dilution of sampled air | 10%[*] | -8%[**] | (-9%[**]) | | – |
| Temperature increase in ref. FT | 5%[***] | -3%[**] | (-5%[**]) | | – |
| $O_x$ formation from OH+NO$_z$ | – | | – | +3%[**] | (+3%[**]) |
| **Conservative sum of biases** | | -44% | (-64%) | +40% | (+59%) |

*from laboratory testing; **from model simulations; ***from estimation

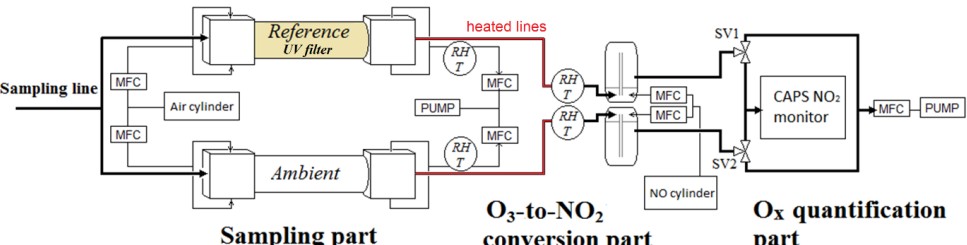

**Figure 1.** Schematic of the OPR instrument. $O_3$ converted into $NO_2$ by reaction with NO. Difference in $O_x$ mixing ratios between the two flow tubes quantified by CAPS. SV: Solenoid Valves. MFC: Mass Flow Controller.





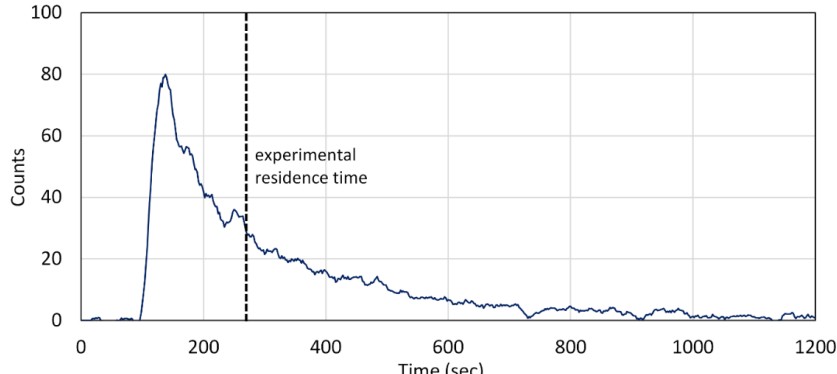

**Figure 2.** Example of pulse experiments for the quantification of the flow tubes residence time. Pulse of toluene generated at the entrance of the flow tube at t=0s.




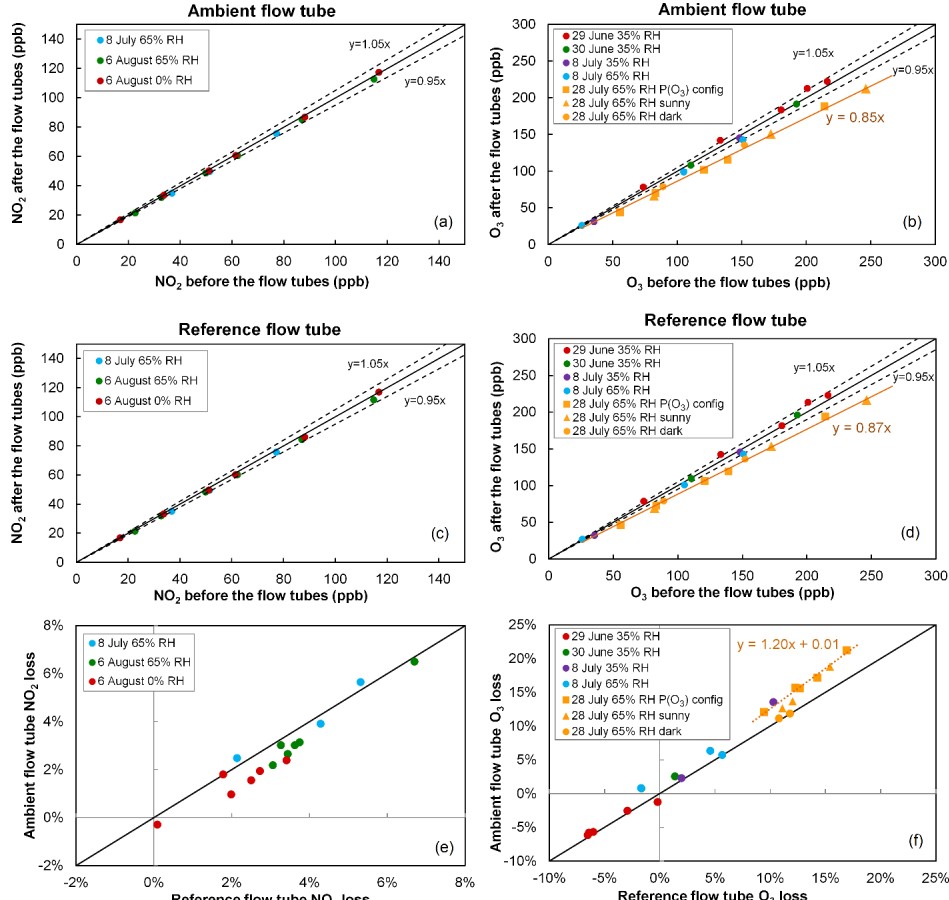

**Figure 3.** $NO_2$ and $O_3$ relative losses measured during the IRRONIC field campaign at different relative humidity values. Losses in the ambient and reference flow tubes are shown in the top and middle panels, respectively. The bottom panel reports the difference in relative losses between the 2 flow tubes. On 28 July $O_3$ losses were measured under sunny conditions (orange squares: ambient flow irradiated and reference flow tube covered by the UV filter; orange triangles: both flow tubes irradiated), and dark conditions (orange circles: both flow tubes covered by an opaque cover).





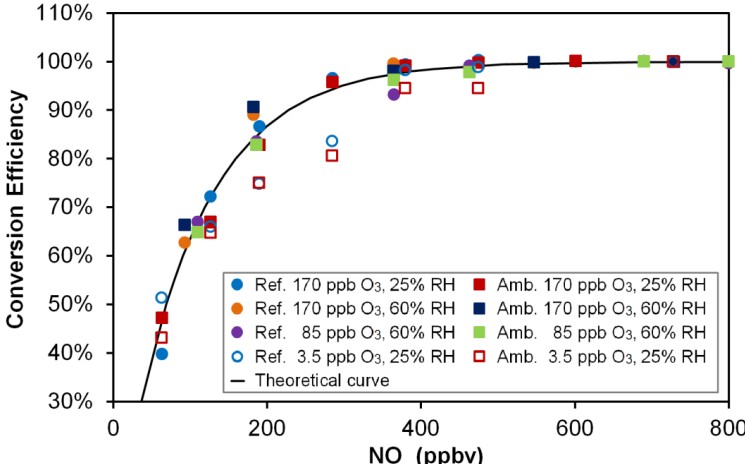

**Figure 4.** $O_3$-to-$NO_2$ conversion efficiency for various NO mixing ratios, $O_x$ levels and relative humidity values. The black curve was calculated from the reaction rate constant between $O_3$ and NO and a reaction time of 23 s. Open symbols (3.5 ppbv $O_3$) are hidden behind the plain symbols for NO>500 ppbv. "Ref." and "Amb." refer to the conversion units coupled to the reference and ambient flow tubes, respectively.



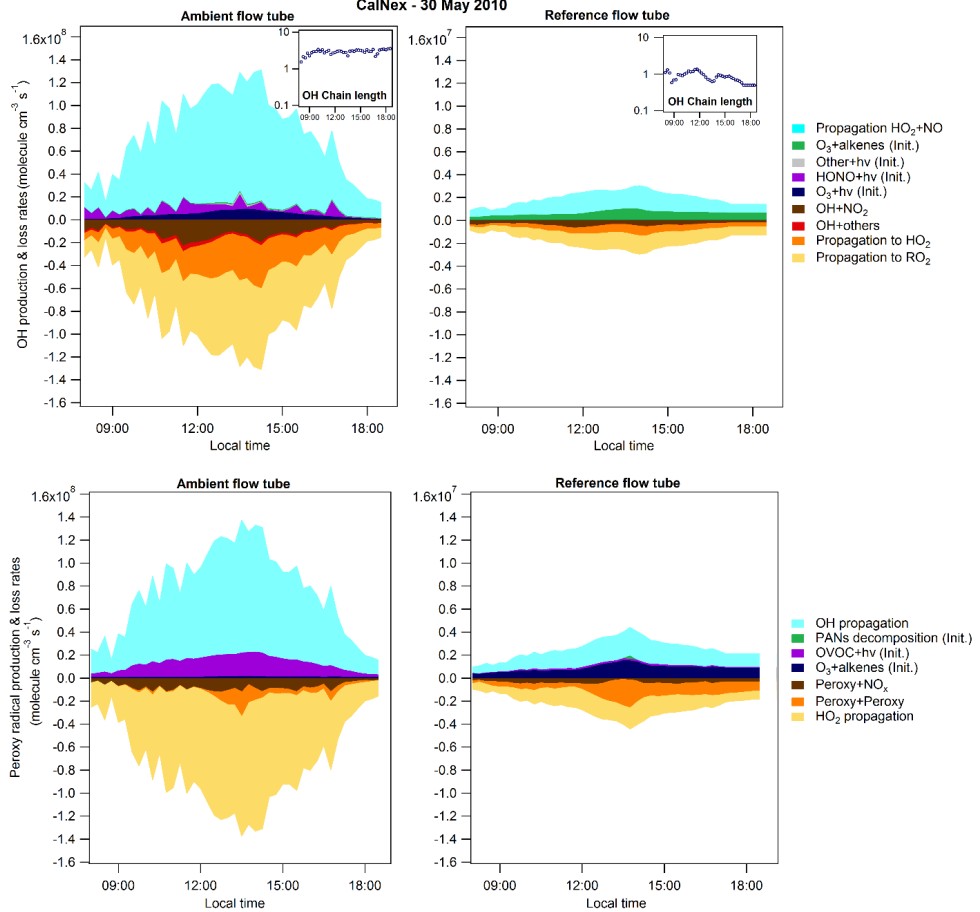

**Figure 5**. OH (top) and total peroxy (HO$_2$+RO$_2$, bottom) radical budgets for 30 May 2010 of the CalNex–2010 campaign. Radical budgets modeled for the ambient (left) and the reference (right) flow tubes. The OH chain length is also presented in an insert (top) for each flow tube. The note (Init.) in the legend indicates initiation reactions.





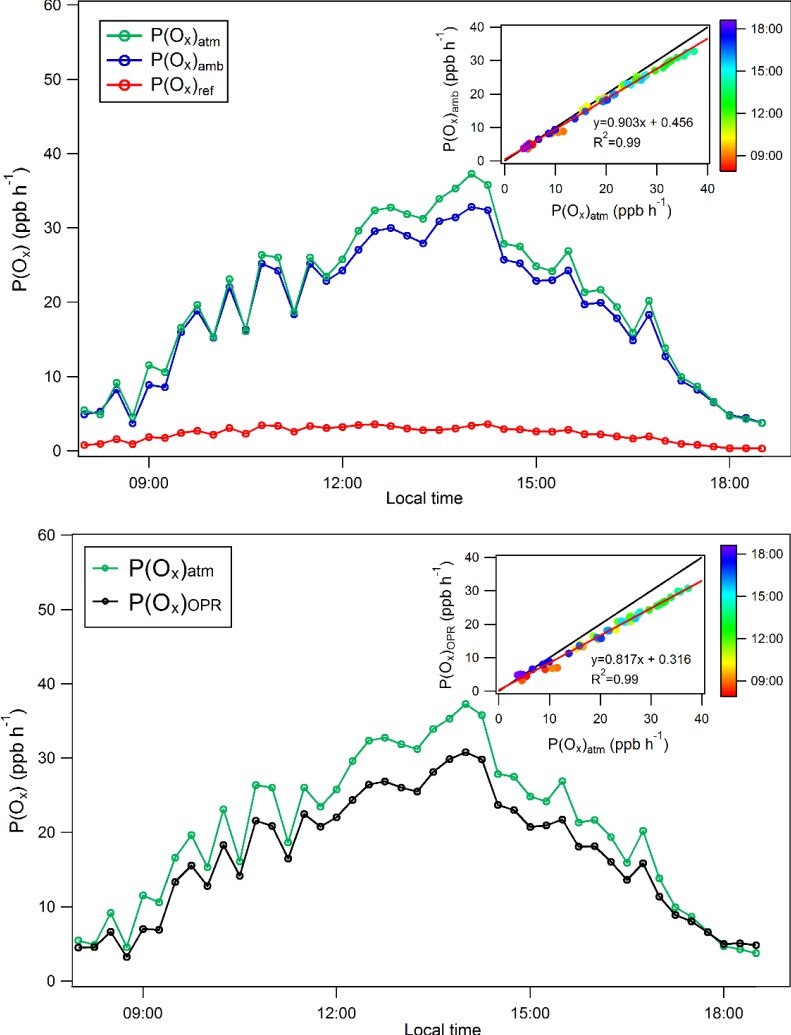

**Figure 6.** Modeling comparison of $P(O_x)$ values. Top: ozone production rates modeled for the atmosphere, $P(O_x)_{atm}$, the ambient flow tube, $P(O_x)_{amb}$, and the reference flow tube, $P(O_x)_{ref}$ for 30 May 2010 of the CalNex–2010 campaign. Bottom: comparison of modeled ozone production rates for the OPR, $P(O_x)_{OPR}$, and the atmosphere, $P(O_x)_{atm}$, for 30 May 2010. Inserts: correlations between $P(O_x)_{atm}$ and $P(O_x)_{amb}$ (top), and $P(O_x)_{atm}$ and $P(O_x)_{OPR}$ (bottom), color-coded by the time of day.



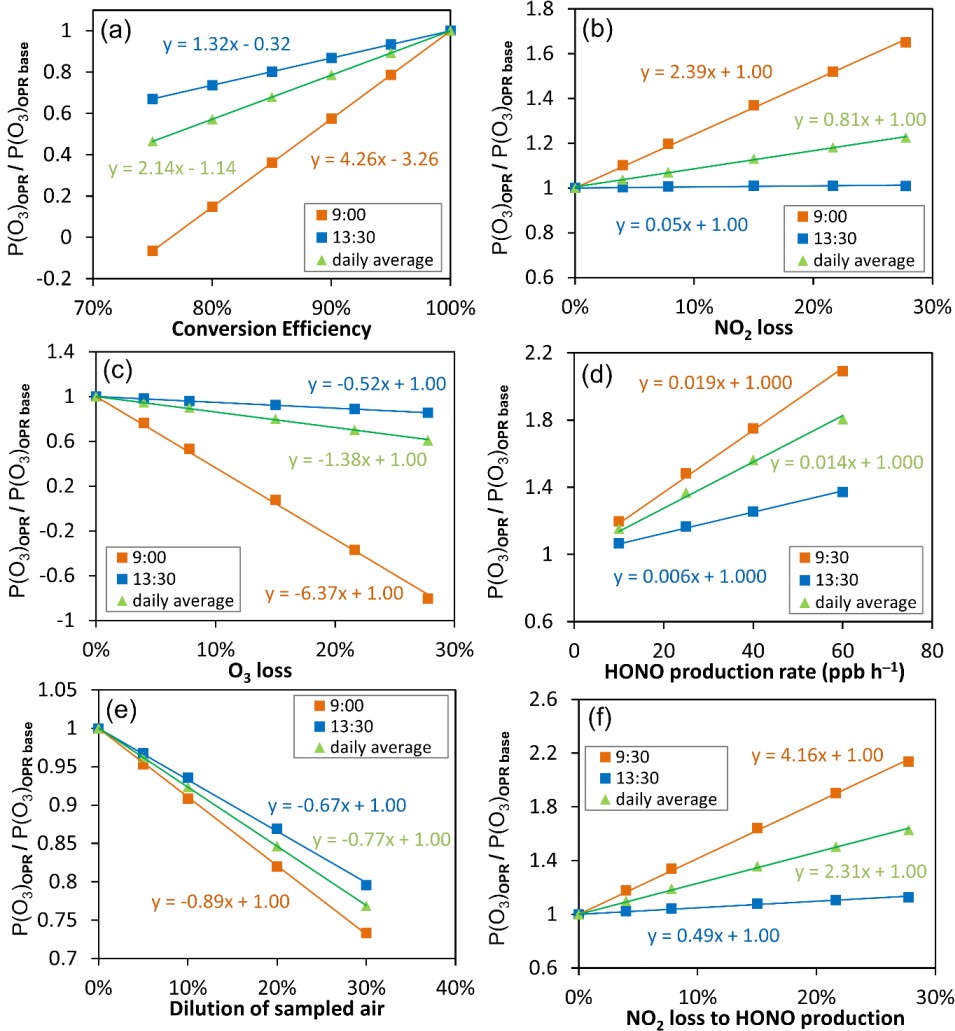

**Figure 7.** Sensitivity tests performed for 30 May 2010 (CalNex-2010) to assess the impact on the P(Ox) measurements of (a) the $O_3$-to-$NO_2$ conversion efficiency, (b) $NO_2$ and (c) $O_3$ dark losses, (d) heterogeneous HONO formation, (e) dilution of ambient air, and (f) $NO_2$ loss towards HONO production in the flow tubes. The results presented here correspond to the two hours of the day identified as lower (blue squares) and upper (orange squares) limits of the impact on the P(Ox) measurements. The daily average behavior is also shown using green triangles.


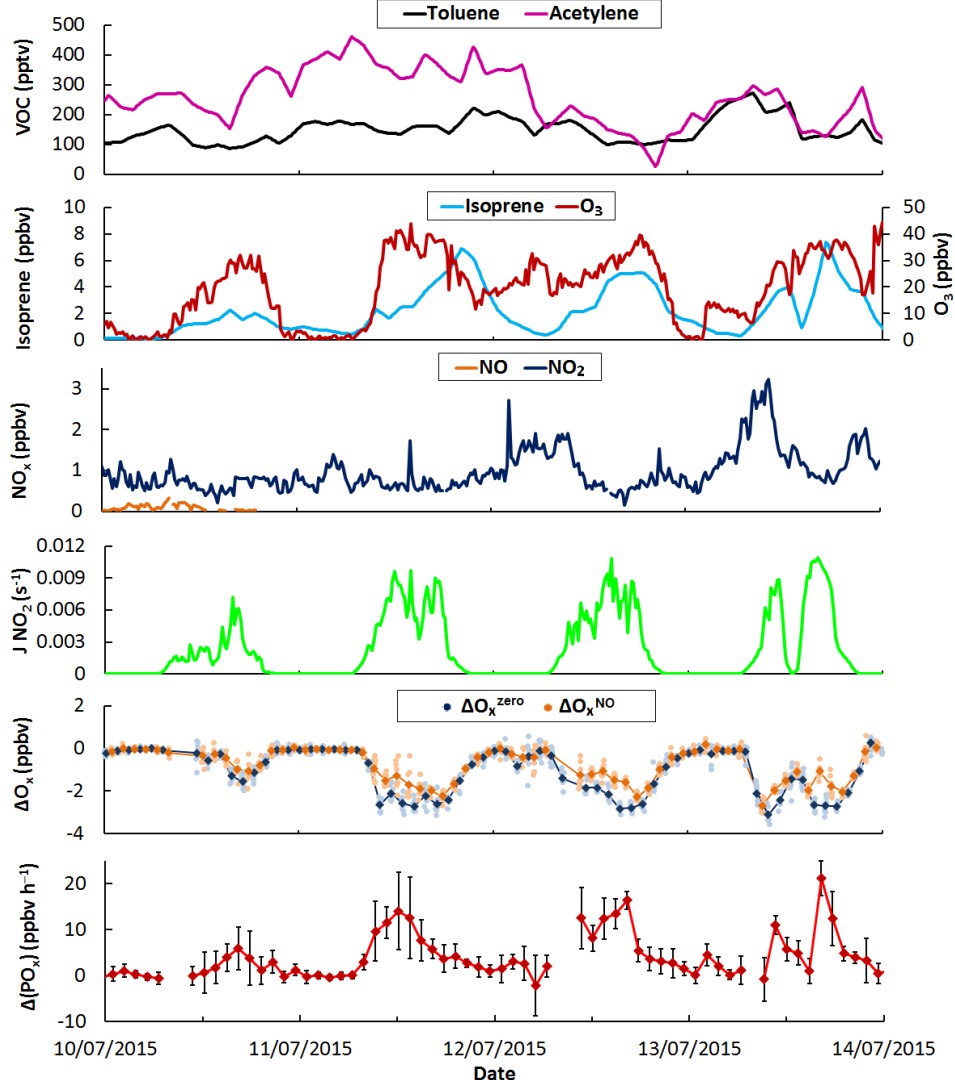

**Figure 8.** Time series of selected trace gases, J(NO$_2$), measured $\Delta$O$_x$ and $\Delta$P(O$_x$) values during four days of the IRRONIC campaign when 6 ppbv of NO was intermittently added in the flow tubes. The light colors on $\Delta$O$_x$ correspond to 2-min measurements while the darker colors are 20-min averaged values. Error bars on $\Delta$P(O$_x$) are 1$\sigma$ on the averaged 20-min measurements.