# Peer review of "Development of an instrument for direct ozone production rate measurements: Measurement"

_Atmospheric Measurement Techniques, 2017_

## Referee Comment (RC1) · Anonymous Referee #1 · 8 Aug 2017

The authors report the investigation of the performance of an instrument measuring ozone production. A very careful characterization of the instrument was done supported by model calculations. Although the conclusion is that potential biases are high so that an application for ambient air measurements is currently not possible, the paper gives valuable information about the difficulties of this measurement. The topic of this work well fits the scope of AMT. Publication in AMT is recommended after addressing the following minor points:

Abstract and later discussion: A clear motivation for the investigation of the OPR sensitivity to NO additions is missing. Some hint in the abstract and further explanation in

the text would help the reader to better understand the purpose of this investigation.

p8: How is the zeroing of the CAPS monitor done? How stable was the zero and what is the uncertainty of the NO2 measurement and the end of the P(Ox) measurement connected to this issue?

p14 l3: Is there an influence from the 4m long heated inlet line on the Ox concentration?

p14 l19: It is not clear what is meant by keeping the NO level constant for days for the NOx additions every 40 min.

p16 l6-8: Is there an influence from nighttime chemistry Ox losses (NO3/N2O5 chemistry) expected in the dark tube compared to the illuminated tube?

p22 / Fig. 5: Why is the NO still relatively high in the dark tube? If this is due to photolysis, why is sunlight not further suppressed? What is the impact of the residual NO on the P(O3) measurement?

Section 3.2.2: I was searching for the assessment of the impact in the sections before. Maybe the authors want to integrate the content of section 3.2.2 in the discussion of the results before.

Technical points:

p2 l5: disturbs instead of disturb

p12 l14: There is a double point after the equation that might not be necessary.

p12 l21: bias instead of Bias

p15 l25: subscript in Ox

p16 l3: subscript in P(Ox)

p17 l11: revealed instead of reveal

p29 l15: subscript in P(Ox)

---

## Referee Comment (RC2) · Anonymous Referee #2 · 26 Aug 2017

This is a review of the scientific article "Development of an instrument for direct ozone production rate measurements: Measurement reliability and current limitations" by Sofia Sklaveniti et al. The paper describes in detail the design of an instrument to detect ozone production as P(Ox). The proposed technique is based on the principle of operation of the MOPS instrument (Cazorla and Brune 2010). The new instrument includes improvements related to sampling and sensitive detection methods. The authors include substantial technical information in regard to the design as well as simulations to assess its performance. They also discuss its advantages as well as caveats and limitations. The paper is well written. However, there are some significant aspects inherent to the development of this research that need to be addressed prior to

potential publication. Below specific comments.

1. Sadanaga et. al. (2017) published an article in which they present an instrument to measure ozone production rates that share very similar strategies as the ones presented by Sklaveniti. The instrument proposed by Sadanaga et. at. is based on the MOPS differential measurement, but uses ozone-to-NO2 conversion followed by NO2 detection with a very sensitive technique. The conversion step consists of adding a large excess of NO to titrate ozone exiting the clear and shaded tubes. This work was published early in 2017. This conversion strategy is the same as the one proposed in the article by Sklaveniti et. al. Another similarity is the material used for the sampling tubes, which in both cases is quartz. However, Sklaveniti et. al. did not cite or discuss the article by Sadanaga. I can speculate that the authors were unaware of the paper by Sadanaga et. al. The ozone conversion strategy is one of the major aspects that would grant novelty to the article by Sklaveniti provided they had published it first. Therefore, academic rigour makes mandatory that Sklaveniti et. al. include a complete section to refer to the work by Sadanaga and discuss similarities and differences with their own instrument. The reference is: Sadanaga et. al., New System for Measuring the Photochemical Ozone Production Rate in the Atmosphere Environ. Sci. Technol., 2017, 51 (5), pp 2871–2878. DOI: 10.1021/acs.est.6b04639

2. I am concerned about the use of quartz for the ambient and reference tubes. Quartz is a material whose surface has chemically active sites prone to adsorption processes. Quartz was not used for the MOPS because significant ozone losses were observed under ambient conditions. The authors do recognize the limitation of using quartz and even acknowledge being unable to zero the instrument when both tubes are exposed to the sun. Nevertheless, it seems to me that the magnitude of the effect of quartz on ozone loss was not thoroughly assessed. The authors present only one test for ozone loss performed under sunny conditions. In addition, Figure 3 shows experimental data with a scale between 0-300 ppbv. I am concerned that the size of the scale is not revealing the true effect of ozone losses. More importantly, the difference in ozone

losses between both tubes could have a significant impact. Additionally, the way the experiment was performed possibly affected the results. For example, if a high concentration of 300 ppbv of ozone was first administered and then concentrations were lowered, this possibly yielded lower losses in magnitude than what potentially could be observed under real conditions. Authors should clarify details about the experimental procedure because high ozone concentrations would have a chemical treatment effect on quartz that ambient concentrations would not cause. From my perspective, the article would benefit if the authors included an evaluation of the effect of ozone losses under real conditions of ozone concentrations and sunlight. As a separate note, the article by Sadanaga also emphasizes in ozone loss tests in dark conditions. This is an additional similarity that needs to be explained.

3. In regard to residence time and flow pattern, the authors discuss their pulse experiment in terms of plug flow and compare their results with the ones for the MOPS. The second version of the MOPS includes substantial improvements to aim for laminar plug flow, so that air molecules reside approximately the same time inside the chambers. However, in the technique proposed by Sklaveniti, a 10% dilution with zero air is applied at the inlet. It seems that at doing so, the authors possibly induced a flow pattern that aims for complete mix as opposed to plug flow. Nevertheless, the discussion is done in terms of plug flow or laminar flow. This apparent contradiction needs to be clarified. Finally, did the authors evaluate a potential flushing effect cause by the zero air on the gases in the central jet? Are measured P(Ox) values similar with and without dilution? It would be appropriate to include some data to demonstrate the benefit of adding dilution to the main flow.

---

## Author Comment (AC1) · 8 Nov 2017

**Response to Anonymous Referee #1**

We are grateful to the first anonymous reviewer for his review and the valuable comments which helped improving our manuscript. For clarity, the comments are reproduced in blue below, followed by our replies.

**Abstract and later discussion: A clear motivation for the investigation of the OPR sensitivity to NO additions is missing. Some hint in the abstract and further explanation in the text would help the reader to better understand the purpose of this investigation.**

As suggested by the reviewer we modified the abstract and the introduction to motivate the investigation of the OPR sensitivity to NO as follows (changes in bold italics):

"... However, an attempt was made to investigate the OPR sensitivity to $NO_x$ by adding NO inside the instrument. This type of investigations ***allows checking whether our understanding of the turnover point between NOx-limited and NOx-saturated regimes of ozone production is well understood and*** does not require measuring ambient OPR but only probing the change in ozone production when NO is added. During IRRONIC, changes in ozone production rates ranging from the limit of detection (3σ) of 6.2 ppbv $h_{-1}$ up to 20 ppbv $h_{-1}$ were observed when 6 ppbv of NO was added into the flow tubes."

We also modified the following paragraph in the introduction section:

"When ozone is produced, reactions of peroxy radicals with NO also lead to the formation of OH, which can then oxidize other molecules of VOCs to produce more peroxy radicals, and as a consequence, more ozone. The propagation chemistry between $RO_x$ (OH, $HO_2$ and $RO_2$) radicals, which fuels ozone production, is terminated either by $NO_x$-$RO_x$ reactions or by cross reactions of $RO_x$ radicals in NO-rich and $NO_x$-poor environments, respectively. These two types of termination reactions lead to different regimes of ozone production referred as $NO_x$-limited or $NO_x$-saturated when the rate of ozone production increases or decreases with $NO_x$, respectively. The turnover point between the two regimes depends on $NO_x$ concentrations, VOC reactivity, and radical production rates (Kleinman, 2005). Since different air quality regulations have to be implemented for the two different regimes, i.e either $NO_x$ or VOC emission regulations, ***investigating the sensitivity of ozone production rates to its precursors during field studies, such as NOx, is important to test our understanding of the turnover point.*** Understanding this complex and non-linear radical chemistry is key for the design of efficient emission control strategies."

*p8: How is the zeroing of the CAPS monitor done? How stable was the zero and what is the uncertainty of the NO2 measurement and the end of the P(Ox) measurement connected to this issue?*

The zero was checked frequently during the field campaign by providing dry zero air to the monitor and was found to change by less than 0.3 ppbv over 12 hours. It is worth noting that a slow drift of the zero does not impact the measurements since the same CAPS monitor was used to measure Ox at the exit of both flow tubes with a switching time of 1 minute. The calculation of $P(O_3)$ implies a subtraction of the measured Ox concentrations, which cancels out any offset in the monitor's zero.

The CAPS monitor was calibrated with a $NO_2$ standard mixture certified at 190±3 ppb (2σ) by LNE (French National Metrology Institute). This uncertainty of 1.5% from the $NO_2$ calibration will propagate in quadrature to $P(O_3)$.

These two points have been clarified in the manuscript in section 2.1.

**p14 l3: Is there an influence from the 4m long heated inlet line on the Ox concentration?**

No tests were performed to check whether the heated lines could lead to a change in Ox concentrations. The rationale behind the use of these lines was to make sure to get the best transmission of Ox species from the flow tubes to the $O_3$-to-$NO_2$ converter and to avoid any condensation of water when the lines are sent inside the laboratory for quantification with the CAPS monitor.

**p14 l19: It is not clear what is meant by keeping the NO level constant for days for the NOx additions every 40 min.**

The NO addition was turned ON and OFF every 40 minutes all along the campaign. The level of NO added in the flow tubes when the addition was turned ON was kept at a constant level for several days. For instance, the flow rate of NO was adjusted to get an addition of 6 ppbv for the time period shown in Figure 8 (10-14 July). The flow was adjusted to other values during other time periods, leading to different mixing ratios of NO. We modified the manuscript to clarify this point.

**p16 l6-8: Is there an influence from nighttime chemistry Ox losses (NO3/N2O5 chemistry) expected in the dark tube compared to the illuminated tube?**

We do not expect a different impact of the $NO_3$/$N_2O_5$ chemistry between the reference and ambient flow tubes during daytime for the reason that the filter covering the tube does not strongly affect $J(NO_3)$. A spectroradiometer (METCON) was used to quantify the change in J-values due to the absorption of the Ultem film. These measurements showed that $J(NO_3)$ was reduced by less than 20%, leading to the conclusion that the photolysis frequency of $NO_3$ in the reference tube was still high enough to keep $NO_3$ at a similar level than in the ambient flow tube, which is negligible during daytime.

**p22 / Fig. 5: Why is the NO still relatively high in the dark tube? If this is due to photolysis, why is sunlight not further suppressed? What is the impact of the residual NO on the P(O3) measurement?**

As shown in Fig. S6, NO mixing ratios range from 10-200 ppt for 30 May after 271 s of residence time in the reference (dark) flow tube. As noted by the reviewer, these mixing ratios are larger than what is expected if ambient NO is titrated by $O_3$ in the flow tube without any other process to reform NO. For this day, NO mixing ratios at the exit of the reference flow tube should range from 0.05-100 ppt when the measured ambient ozone mixing ratios are considered. As mentioned above, the

transmission of the Ultem filter was characterized using a spectroradiometer. These measurements showed that $J(NO_2)$ in the reference flow tube is approximately 2% of ambient $J(NO_2)$, which is sufficient to convert a fraction of $NO_2$ into NO, leading to higher NO mixing ratios than expected.

This residual NO contributes to ozone formation in the reference flow tube as shown in Figure 6 for 30 May.

*Section 3.2.2: I was searching for the assessment of the impact in the sections before. Maybe the authors want to integrate the content of section 3.2.2 in the discussion of the results before.*

We thank the reviewer for this suggestion. However, including the content of section 3.2.2 in the previous discussion will make the discussion more complex and we decided to keep section 3.2.2 as is for the sake of clarity.

*Technical points: p2 l5: disturbs instead of disturb ; p12 l14: There is a double point after the equation that might not be necessary ; p12 l21: bias instead of Bias ; p15 l25: subscript in Ox ; p16 l3: subscript in P(Ox) ; p17 l11: revealed instead of reveal ; p29 l15: subscript in P(Ox)*

These corrections have been done.

---

## Author Comment (AC2) · 8 Nov 2017

**Response to Anonymous Referee #2**

We are grateful to the second anonymous reviewer for his review and the valuable comments which helped improving our manuscript. For clarity, the comments are reproduced below in blue, followed by our replies.

*Sadanaga et. al. (2017) published an article in which they present an instrument to measure ozone production rates that share very similar strategies as the ones presented by Sklaveniti. The instrument proposed by Sadanaga et. at. is based on the MOPS differential measurement, but uses ozone-to-NO2 conversion followed by NO2 detection with a very sensitive technique. The conversion step consists of adding a large excess of NO to titrate ozone exiting the clear and shaded tubes. This work was published early in 2017. This conversion strategy is the same as the one proposed in the article by Sklaveniti et. al. Another similarity is the material used for the sampling tubes, which in both cases is quartz. However, Sklaveniti et. al. did not cite or discuss the article by Sadanaga. I can speculate that the authors were unaware of the paper by Sadanaga et. al. The ozone conversion strategy is one of the major aspects that would grant novelty to the article by Sklaveniti provided they had published it first. Therefore, academic rigour makes mandatory that Sklaveniti et. al. include a complete section to refer to the work by Sadanaga and discuss similarities and differences with their own instrument. The reference is: Sadanaga et. al., New System for Measuring the Photochemical Ozone Production Rate in the Atmosphere Environ. Sci. Technol., 2017, 51 (5), pp 2871–2878. DOI: 10.1021/acs.est.6b04639*

We thank the reviewer to bring this publication to our attention. A paragraph has been added in the introduction to acknowledge this work.

*"A recent publication from Sadanaga et al. (2017) also reports the development and the field deployment of an instrument to measure ozone production rates. The main differences with MOPS is the use of two quartz flow tubes instead of Teflon chambers, an $O_3$-to-$NO_2$ conversion unit, and a $NO_2$ detection by laser-induced fluorescence. While quartz was chosen for the flow tubes, their inner surface is covered by a Teflon film. The reported detection limit is 0.5 ppbv h$^{-1}$ for 60-s measurements. $P(O_3)$ values ranging from the detection limit up to 11 ppbv h$^{-1}$ were reported for three days of measurements in a forested area characterized by low mixing ratios of $O_3$ (<10 ppbv) and $NO_x$ (< 1ppbv). "*

A discussion has also been added in section 3.4, whose title has been modified to reflect the new content.

*"3.4 Comparison to previously published apparatus and potential improvements for the OPR instrument*

*......*

*The instrument design reported by Sadanaga et al. (2017) does not seem to be impacted by a photolytic loss of ozone on the quartz flow tubes whose inner surface was coated with Teflon. Interestingly, these authors report dark losses of ozone on the order of 8-10% on the uncoated quartz surface for a residence time of 21 minutes in the tubes, which are consistent with the*

*reported dark  loss of less than 5% observed in our study for $O_3$-conditioned flow tubes and a residence time of 4.5 minutes . The Teflon coating seems to remove or to reduce the photolytic loss of ozone to a negligible level on this instrument."*

**I am concerned about the use of quartz for the ambient and reference tubes. Quartz is a material whose surface has chemically active sites prone to adsorption processes. Quartz was not used for the MOPS because significant ozone losses were observed under ambient conditions. The authors do recognize the limitation of using quartz and even acknowledge being unable to zero the instrument when both tubes are exposed to the sun. Nevertheless, it seems to me that the magnitude of the effect of quartz on ozone loss was not thoroughly assessed. The authors present only one test for ozone loss performed under sunny conditions. In addition, Figure 3 shows experimental data with a scale between 0-300 ppbv. I am concerned that the size of the scale is not revealing the true effect of ozone losses. More importantly, the difference in ozone losses between both tubes could have a significant impact. Additionally, the way the experiment was performed possibly affected the results. For example, if a high concentration of 300 ppbv of ozone was first administered and then concentrations were lowered, this possibly yielded lower losses in magnitude than what potentially could be observed under real conditions. Authors should clarify details about the experimental procedure because high ozone concentrations would have a chemical treatment effect on quartz that ambient concentrations would not cause. From my perspective, the article would benefit if the authors included an evaluation of the effect of ozone losses under real conditions of ozone concentrations and sunlight. As a separate note, the article by Sadanaga also emphasizes in ozone loss tests in dark conditions. This is an additional similarity that needs to be explained.**

As the reviewer mentioned it, we did acknowledge the measurement limitations due to the use of quartz. The dark and photo-enhanced loss of ozone at the inner surface of the flow tubes was extensively studied for this OPR instrument. In this publication, we only showed loss tests performed under dark conditions and when the flow tubes were irradiated during IRONIC since other results from this campaign are also presented. However, multiple tests were performed before and after the campaign under different conditions of illumination, RH, and ozone mixing ratios to extensively investigate the loss of ozone on the quartz material.

We present a few of these tests below. The results shown in Figure R1 were obtained after flowing 200 ppbv of $O_3$ in the flow tubes for two days at a relative humidity of 60%. Dark ozone loss measurements were performed at relative humidity values of 45% and 70% (approx. 21°C) for a range of ozone mixing ratios similar to that observed in the troposphere (10-110 ppbv) . These tests show that the absolute loss scales linearly with the ozone mixing ratio and indicate an average relative loss of 2.9% and 1.8% for the ambient and reference flow tubes, respectively, including both relative humidity values that were examined. These values are similar to that observed during at least the first 10 days of ambient measurements during the IRRONIC campaign (Figure 3). It should be noted that the tubes were also conditioned with ozone (160 ppbv) over 12 hours before starting ambient measurement during IRRONIC.

[Figure]

**Figure R1:** $O_3$ loss tests performed in the laboratory for the ambient (left) and the reference (right) flow tubes. The abbreviation "FTs" indicates "flow tubes".

For the experiments shown in Figure R1, it is interesting to note that measurements performed before flowing a concentrated mixture of $O_3$ in the tubes had highlighted elevated losses on the order of 20%. This large loss may be due a contamination of the surface with unsaturated organic species as mentioned in the manuscript to explain the increase of the dark loss during the IRRONIC campaign. However, as pointed out by the reviewer, flowing a large ozone concentration in the flow tubes for cleaning purposes may have also led to a chemical treatment of the surface, which in turn may have led to a decrease of the dark loss. This point will be acknowledged in the publication and will be mentioned as a potential reason for the increase of the dark loss observed during IRRONIC.

From the tests shown in Figure R1 and in Figure 3 for IRRONIC, it appears that the difference in dark loss between the two flow tubes is lower than 1%. Considering an ambient ozone mixing ratio of 50 ppbv, which is close to the highest level observed during IRONIC, a 1% higher loss in the ambient flow tube would lead to bias of approximately -7 ppb/h in P($O_3$).

Additional tests were performed after the IRRONIC campaign to investigate the photolytic loss of $O_3$ on the quartz material. These tests were performed by irradiating the flow tubes with UV lamps (312 and 365 nm), introducing known mixing ratios of ozone in the flow tubes and varying the humidity or light conditions. An example of tests where an $O_3$ mixing ratio of 55 ppbv was used is shown in Figure R2.

Initially, the ozone loss was quantified under dark conditions (blue circles). Then the lamps were turned ON (green circles) using different settings: (i) all lamps ON (4 × 312 nm & 4 × 365nm), (ii) side A ON (2 × 312 nm & 2 × 365 nm, lamps located above ambient flow tube), (iii) side B ON (2 × 312 nm & 2× 365 nm, lamps located above reference flow tube), (iv) 2 × 365 nm lamps ON. These steps were repeated for two or three different humidity values for each experiment. The objective was to investigate the humidity dependence of the dark and photo-enhanced ozone losses, as well as investigating the wavelengths that mostly contribute to the photo-enhanced loss in the ambient flow tube.

[Figure]

**Figure R2:** Relative ozone loss in each flow tube as a function of absolute humidity. The blue and green circles indicate losses under dark and irradiated conditions, respectively. The difference between these conditions is shown using the red circles.

Figure R2 shows the relative ozone loss in each flow tube as a function of absolute humidity during field testing performed on the campus of Birmingham, after four days of outdoor measurements, and without conditioning the tubes with $O_3$. It is interesting to note that a large dark loss of approximately 10-20% was observed in the flow tubes, indicating either a contamination of the flow tubes with unsaturated species or a larger loss rate on an unconditioned quartz surface. In addition, the dark loss observed during these tests was dependent on humidity, which contrasts to that observed during IRRONIC.

The green markers indicate the total loss (dark + photolytic) observed when the lamps were turned ON. Labels on the green markers describe the different irradiation conditions. Assuming that the dark loss does not change when the lamps are turned ON, the difference between the total loss and the dark loss yields the photolytic component of the loss, which is shown using red markers in the bottom panels. For the ambient flow tube, when all lamps are turned ON (for the two lower humidity conditions), the photolytic ozone loss is approximately 7.5%. Turning OFF half the lamps also leads to a decrease of the photolytic loss by half, indicating that the photolytic loss may depend linearly on J-values. In addition, it is clear from this figure that short wavelengths close to 312 nm are causing the photo-enhanced loss since the photolytic loss is reduced to an insignificant level when the 365 nm lamps are used alone.

All the tests discussed above unambiguously show that the ozone loss at the quartz surface is strongly dependent on the irradiation reaching the surface. This issue is acknowledged in the publication and is shown to be the main limitation on this OPR instrument. From the reviewer comment, it also appears that the loss may be dependent on whether the surface was conditioned with ozone before ambient measurements and this issue will also be acknowledged in the revised publication.

A paragraph has been added at the end of section 3.1.2:

*"Additional tests were performed after the campaign under different conditions of illumination, RH, and ozone mixing ratios to thoroughly investigate the loss of ozone on the quartz material. Overall, these tests showed that the dark loss can be reduced below 5% for several days of ambient measurements if the quartz flow tubes are conditioned with elevated $O_3$ mixing ratios at high relative humidity. These results indicate that the low value observed for the loss after the conditioning period may be due to (i) a clean-up of the surfaces, removing unsaturated organic species that may be absorbed on the quartz surface, or (ii) a chemical treatment of the surface, deactivating sites where ozone could be lost during ambient measurements. Tests were also performed to investigate the potential photo-enhanced loss of ozone discussed above. These tests were performed by irradiating the two flow tubes with UV lamps (312 and 365 nm), introducing known mixtures of ozone/zero air in the flow tubes and varying humidity and/or light conditions. While a photo-enhanced loss of ozone was not observed in the reference flow tube covered with the UV filter, a significant photo-enhanced loss of up to 7.5% was observed for the ambient flow tube when the 312 nm lamps were used, with a dependence on light intensity. In contrast, irradiating the ambient flow tube with the 365 nm lamps did not lead to a photo-enhanced loss, indicating that lower wavelengths are inducing the loss process responsible of the photo-enhanced loss. This issue is further discussed in the field deployment section (3.3)."*

The comparison of ozone loss rates between this instrument and the one described in Sadanaga et al. (2017) has been addressed for the previous comment.

**In regard to residence time and flow pattern, the authors discuss their pulse experiment in terms of plug flow and compare their results with the ones for the MOPS. The second version of the MOPS includes substantial improvements to aim for laminar plug flow, so that air molecules reside approximately the same time inside the chambers. However, in the technique proposed by Sklaveniti, a 10% dilution with zero air is applied at the inlet. It seems that at doing so, the authors possibly induced a flow pattern that aims for complete mix as opposed to plug flow. Nevertheless, the discussion is done in terms of plug flow or laminar flow. This apparent contradiction needs to be clarified.**

It was not our intention to discuss the pulse experiment in terms of plug flow. As mentioned in section 3.1.1, the pulse observed experimentally is treated as a probability distribution, with the average residence time in the flow tubes being the mean of the probability distribution. The comparison of the measured residence time to that expected from plug flow conditions may be confusing as written in the publication. We will modify the text as follows to clarify that we don't consider plug flow conditions in the flow tubes.

"As described in the experimental section, pulses of toluene were injected in the flow tubes to quantify the mean residence time. One of the 5 experiments that were conducted is shown in Figure 2. The pulse shape is asymmetric and exhibits a long tail, indicating that a large range of residence times is observed in the flow tubes. The toluene pulse is treated as a probability distribution of the time variable $t$, with the average residence time in the flow tubes being the mean of the probability distribution. The latter is calculated as a weighted average of the possible values that the time variable can take. The average residence time from the 5 toluene pulse experiments was 4.52 ± 0.22 min ($1\sigma$). The uncertainty reported for the residence time will lead to a 4.9% error ($1\sigma$) on the $P(O_x)$ measurements. ***While plug flow conditions are not met in the flow tubes, it is interesting to note that a residence time of 4.79 min would be expected from plug flow conditions at a total flow rate of 2.25 L min$^{-1}$ for a volume of 10.8 L in each flow tube.*** The asymmetry of the peak indicates that the flow rate at the central axis of the tube is larger, with the first molecules of toluene being sampled after approximately 2 minutes (Fig. 2). These observations are similar to that reported by Cazorla and Brune (2010) for sampling chambers exhibiting a different geometry and operated under different flow conditions. ***A similar asymmetric shape is observed for the pulse.*** Further work is needed on the OPR instrument to reduce the skewness of the time distribution."

**Finally, did the authors evaluate a potential flushing effect cause by the zero air on the gases in the central jet? Are measured P(Ox) values similar with and without dilution? It would be appropriate to include some data to demonstrate the benefit of adding dilution to the main flow.**

The effect of an addition of zero air at the periphery of the internal inlet was tested using fluid dynamic calculations. These calculations showed that adding zero air helps reducing recirculation eddies near the inlet. Examples of simulations performed for different flow tube geometries are shown in Figure R3, panel c being the geometry used for the OPR instrument. On the left side of this Figure, the geometry of the flow tube is shown in opaque and transparent forms, while on the right side the streamlines of the flow are color coded with the flow velocity. The streamlines describe lines that are tangential to the instantaneous velocity direction and show the direction in which a massless fluid particle will travel at any point in time. In all cases, the flow is entering the tube from the left side, and exiting from the right side. The different geometries and their impact on the flow pattern are described below to emphasize the advantages of the geometry used for the OPR instrument.

The first geometry (a) consists of a simple cylindrical tube, with inlet and outlet outer diameters of 2.54 cm (1") and 1.27 cm (½"), respectively. As shown on the right side of the figure, the streamlines form recirculation eddies all along the length of the flow tube. For this geometry, a large fraction of the air entering the tube is in contact with the walls for a long time and is sampled by the monitor. This recirculation effect is undesirable, since wall contacts are amplified. These eddies result from the sudden increase of the cross section of the tube near the inlet, often referred to as backward facing step. The size of these eddies depend on the geometry and the flow velocity, generally with larger and more intense eddies for higher flow rates and higher Reynolds numbers.

The second geometry (b) includes a curved conical inlet that smoothen the backward facing step. The length of this inlet is 20 cm, and the cross section increases from 2.54 cm (1") to 14 cm. It has been shown that the best flow pattern is achieved when a small diverging angle is implemented (5 to 12°, depending on the Reynolds number). In our case, in order to keep a reasonable length for the flow

tube, it is not possible to use an angle smaller than 35°. This inlet geometry is expected to reduce the recirculation issues but the flow separation is unavoidable. On the outlet side, the air is sampled at 0.75 L/min through an internal outlet (Ø=1.27 cm, ½") that is located at the center (radial position) of the flow tube. The purpose is to sample air coming along the central axis of the flow tube that has interacted less with the walls, while the air in contact with the walls is extracted using an external pump at 1.5 L/min on the outer periphery of the outlet. A close inspection of the streamlines of this geometry shows that eddies are reduced to the first 45 cm of the tube. Geometry (b) is an improvement compared to geometry (a), regarding the extent of the recirculation eddies and the sampling taking place only at the central axis of the tube.

The third geometry (c) has a similar sampling collector (inner outlet), but using a conical shape to reduce the perturbation of the flow near the area of the outlet. The internal outlet starts from an outer diameter of 3 cm at the point where the flow is sampled, decreasing to a diameter of 1.27 (½") cm after 10 cm. The flow rates at the outlet are the same as in geometry (b). At the inlet side, ambient air is sampled by a curved conical internal inlet at the center of the flow tube, while zero air is injected at 0.25 L/min at the periphery of the inlet inside the flow tube. The injection flow rate was limited to approximately 10% of the total flow rate, to minimize the impact of a dilution on $P(O_3)$ measurements. This additional air helps keeping the flow forward, minimizing recirculation eddies, and therefore reducing the impact of the walls on the chemical composition of the sample. The internal inlet has an initial inner diameter of 2.2 cm that increases to 7 cm over a length of 20 cm, until the point where the flow enters the cylindrical flow tube, leading to an entrance angle of 11.4°. As can be seen from the streamlines, the recirculation eddies are minimized to the first 30 cm of the flow tube, resulting to a clear improvement compared to geometries (a) and (b).

[Figure]

**Figure R3:** Computational Fluid Dynamics simulations of various flow tubes – on the left, the geometry in opaque and transparent form with the flow rate boundary conditions, and on the right the streamlines.

We did not investigate the effect of the addition of zero air on the $P(O_3)$ measurements through laboratory experiments or field testing. This effect was studied through the modeling work reported in section 3.2.2 assuming a 10% dilution of the sampled air. The simulations showed that an underestimation of up to 9% could arise from dilution.